# USP9X regulates centrosome duplication and promotes breast carcinogenesis

Xin Li[1,*], Nan Song[1,*], Ling Liu[1], Xinhua Liu[1], Xiang Ding[2], Xin Song[3], Shangda Yang[1], Lin Shan[1], Xing Zhou[1], Dongxue Su[1], Yue Wang[1], Qi Zhang[1], Cheng Cao[1], Shuai Ma[1], Na Yu[1], Fuquan Yang[2], Yan Wang[1], Zhi Yao[4], Yongfeng Shang[1,5] & Lei Shi[1,4]

Defective centrosome duplication is implicated in microcephaly and primordial dwarfism as well as various ciliopathies and cancers. Yet, how the centrosome biogenesis is regulated remains poorly understood. Here we report that the X-linked deubiquitinase USP9X is physically associated with centriolar satellite protein CEP131, thereby stabilizing CEP131 through its deubiquitinase activity. We demonstrate that USP9X is an integral component of centrosome and is required for centrosome biogenesis. Loss-of-function of USP9X impairs centrosome duplication and gain-of-function of USP9X promotes centrosome amplification and chromosome instability. Significantly, USP9X is overexpressed in breast carcinomas, and its level of expression is correlated with that of CEP131 and higher histologic grades of breast cancer. Indeed, USP9X, through regulation of CEP131 abundance, promotes breast carcinogenesis. Our experiments identify USP9X as an important regulator of centrosome biogenesis and uncover a critical role for USP9X/CEP131 in breast carcinogenesis, supporting the pursuit of USP9X/CEP131 as potential targets for breast cancer intervention.

[1] 2011 Collaborative Innovation Center of Tianjin for Medical Epigenetics, Tianjin Key Laboratory of Medical Epigenetics, Department of Biochemistry and Molecular Biology, School of Basic Medical Sciences, Tianjin Medical University, Tianjin 300070, China. [2] Laboratory of Proteomics, Institute of Biophysics, Chinese Academy of Sciences, Beijing 100101, China. [3] Research Center of Basic Medical Science, Tianjin Medical University, Tianjin 300070, China. [4] Tianjin Key Laboratory of Cellular and Molecular Immunology, Key Laboratory of Immune Microenvironment and Disease (Ministry of Education), Department of Immunology, School of Basic Medical Sciences, Tianjin Medical University, Tianjin 300070, China. [5] Key Laboratory of Carcinogenesis and Translational Research (Ministry of Education), Department of Biochemistry and Molecular Biology, School of Basic Medical Sciences, Peking University Health Science Center, Beijing 100191, China. * These authors contributed equally to this work. Correspondence and requests for materials should be addressed to Y.S. (email: yshang@hsc.pku.edu.cn) or to Lei.S. (email: shilei@tmu.edu.cn).

Centrosome, a non-membrane organelle in most mammalian cells, is the primary site of microtubule nucleation and organization during interphase and mitosis in diploid cells[1]. Composed of two orthogonally positioned cylindrical centrioles surrounded by clusters of small granular structures termed centriolar satellites, centrosome is important for cell division, polarity, growth and migration[2]. Similar to DNA, centrosome duplicates once per cell cycle, which is initiated around the time of S phase entry and completed by the end of $G_2$ phase[3]. Acquisition of more than two centrosomes (centrosome amplification) severely disturbs mitotic process and cytokinesis via formation of more than two spindle poles, resulting in an increased frequency of chromosome segregation errors (chromosome instability)[4]. Indeed, centrosome amplification has been frequently observed in various types of cancers and believed to be associated with cancer development or progression[5–8]. However, the molecular mechanism by which centrosome duplication is regulated and how centrosome amplification is introduced and contributes to cancer development/progression are still poorly understood.

Centrosome-associated CEP family proteins are the active component of centrosome and are believed to play important roles in the biogenesis and functionality of centrosome[6,9,10]. It has been proposed that CEP protein-coding genes represent potent tumour suppressors or oncogenes[7]. Among the CEP proteins, CEP131 (also termed azacytidine-inducible-1, AZI1) is an evolutionarily conserved centriolar satellite protein required for cilia formation[10–13]. Interestingly, proteomics analysis and molecular study suggest that CEP131 is a ubiquitinated protein[14,15], and, significantly, CEP131 has been reported to play an important role in the maintenance of the genome stability at the time of cell cycle progression[13], suggesting that CEP131 is required for proper centrosome duplication and hinting a potential role for this protein in cancer development and progression.

Protein ubiquitination is a reversible reaction that is constantly opposed by deubiquitination, exemplified by the existence of a large family of deubiquitinating enzymes (DUBs)[16]. The human genome encodes ~95 putative DUBs[17]. Among these DUBs, the ubiquitin-specific peptidase 9, X-linked (USP9X, also known as FAM for Drosophila fat facets in mouse[18]), has been reported to target several cytosolic proteins and to regulate multiple cellular activities including protein trafficking/endocytosis[19], polarity[20], apoptosis and death[21], cell growth and migration[22,23], immune response[24,25], neurogenesis[26] and autophagy[27,28]. In addition, USP9X has been implicated in several pathological states including Turner syndrome[29], X-linked intellectual disability[22], seizures[30], Parkinson's disease[28] and various types of malignancies[31–33]. However, mechanistic insights into the role of USP9X in cancer development and progression remain to be investigated.

In this study, we report that USP9X is an integral component of centrosome and is required for centrosome biogenesis through stabilizing the centriolar satellite protein CEP131. We demonstrate that dysregulation of USP9X contributes to centrosome amplification, chromosome instability and breast carcinogenesis.

## Results

**USP9X is physically associated with CEP131.** In order to further understand the biological function of USP9X, we employed affinity purification and mass spectrometry to interrogate USP9X interactome. Specifically, whole-cell extracts from HeLa cells with doxycycline (Dox)-inducible expression of stably integrated FLAG-USP9X were prepared and subjected to affinity purification and mass spectrometry analysis (Fig. 1a). The results

revealed that USP9X is associated with multiple proteins including itchy E3 ubiquitin protein ligase (ITCH), an ubiquitin ligase known to be associated with USP9X (ref. 34). Interestingly, CEP131, a centriolar satellite protein, was also identified in the USP9X-containing protein complex (Fig. 1a and Supplementary Data 1).

To confirm the association of USP9X with CEP131, co-immunoprecipitation experiments were performed with HeLa cell extracts and the results showed that USP9X was efficiently co-immunoprecipitated with CEP131, but not with another centrosomal protein CP110 (ref. 35), although USP33, another DUB protein, could be effectively co-immunoprecipitated with CP110 (ref. 36; Fig. 1b). Reciprocally, CEP131 was efficiently co-immunoprecipitated with USP9X, but not with USP33, although CP110 could be efficiently co-immunoprecipitated with USP33 (Fig. 1b). Similar observations were also detected in MCF-7 cells, HEK293T cells and U2OS cells (Fig. 1b). The association of USP9X with other interactors identified in mass spectrometry analysis was also validated by co-immunoprecipitation assays (Supplementary Fig. 1a).

We next generated a HeLa cell line with Dox-inducible expression of stably integrated FLAG-CEP131. Immunopurification of CEP131 from HeLa cell extracts with anti-FLAG and analysis of the CEP131-containing protein complex by mass spectrometry revealed that CEP131 was associated with PCM1 (pericentriolar material 1) and several other proteins (Fig. 1c and Supplementary Data 1). Remarkably, USP9X was also detected as a CEP131-interacting protein in these experiments (Fig. 1c).

To further validate the interaction between USP9X and CEP131, protein fractionation experiments were carried out by fast protein liquid chromatography (FPLC) with Superose 6 column and a size exclusion approach. The results indicate that native CEP131 from HeLa cells was eluted with an apparent molecular mass much greater than that of the monomeric protein, and that the elution pattern of CEP131 was largely overlapped with that of USP9X (Fig. 1d). Furthermore, analysis of the FLAG-CEP131 affinity eluate by FPLC with Superose 6 gel filtration revealed that the majority of purified FLAG-CEP131 existed in a multiprotein complex, which peaked in fraction 17 containing PCM1 and USP9X, but not USP33 (Supplementary Fig. 1b).

To further consolidate the interaction between USP9X and CEP131 and to gain insights into the molecular detail involved in the interaction between these two proteins, FLAG-tagged domain deletion mutants of CEP131 were generated and transfected into HeLa cells. Co-immunoprecipitation analysis demonstrated that the N-terminal region of CEP131 upstream of the first coiled-coil domain is required for the interaction of CEP131 with USP9X (Fig. 1e, upper panel). Similarly, domain mapping of the molecular interface of USP9X required for CEP131 binding revealed that a sequence spanning amino acid 611 to 1,553 in the middle region of USP9X (USP9X/M) surrounded by the N-terminal α–α superhelix and the ubiquitin-specific peptidase domain was required for the interaction of UPS9X with CEP131 (Fig. 1e, lower panel). In addition, pull-down experiments with bacterially expressed glutathione S-transferase (GST)-CEP131 and Sf9 cell-purified His-tagged USP9X truncation mutants revealed that USP9X/M directly interacts with CEP131 (Fig. 1f). Moreover, we demonstrated CEP131 N-terminal region directly interacts with USP9X/M (Fig. 1g). Collectively, these experiments revealed a molecular interface between USP9X and CEP131 in which the middle region of USP9X interacts with the N-terminal domain of CEP131.

**USP9X is co-localized with CEP131 in centrosome.** The physical interaction of deubiquitinase USP9X with centrosomal

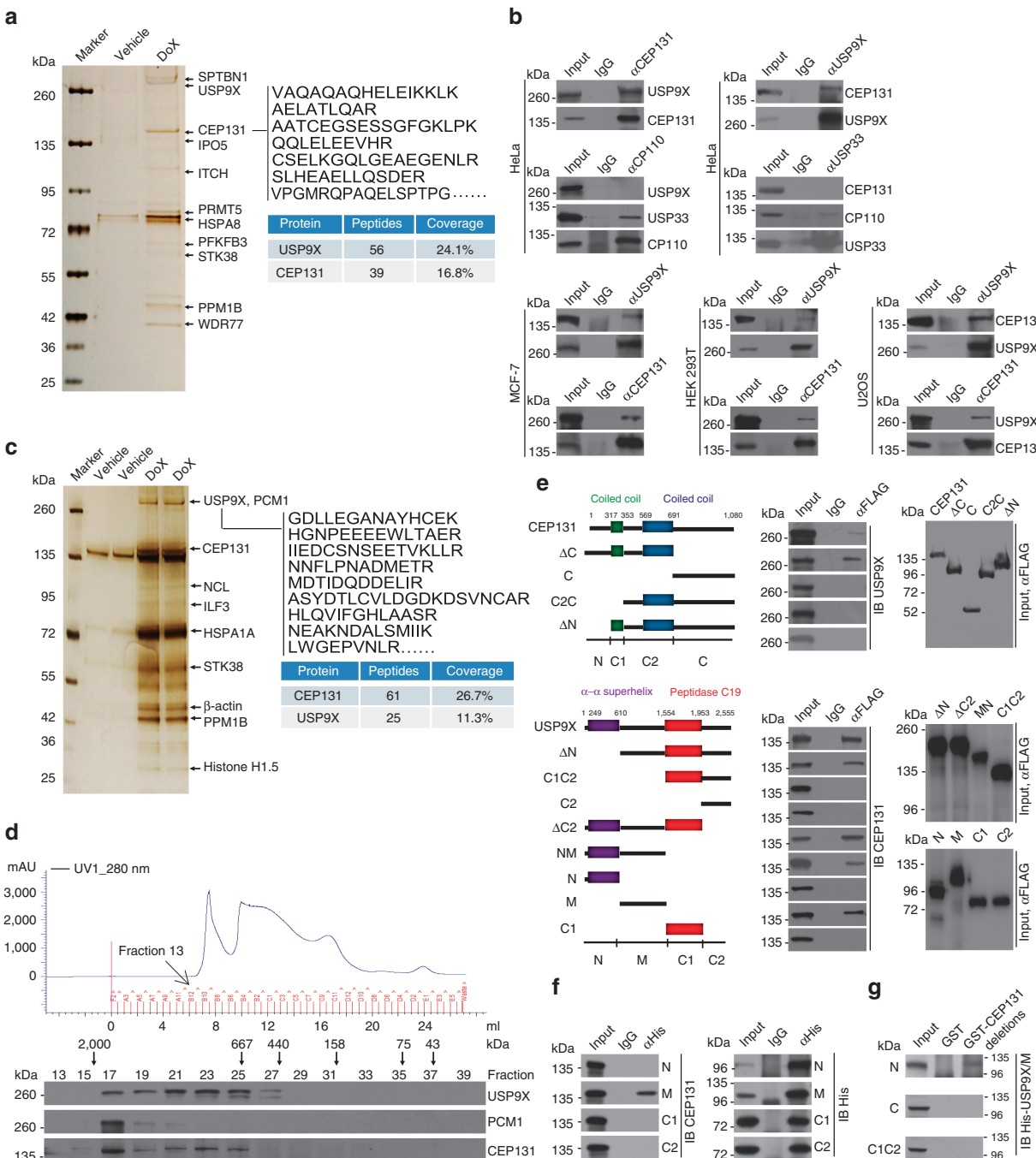

**Figure 1 | Deubiquitinase USP9X is physically associated with the centriolar satellite protein CEP131.** (**a**) Immunoaffinity purification of USP9X-containing protein complexes. Whole-cell extracts from HeLa cells with Dox-inducible expression of stably integrated FLAG-USP9X were purified with an anti-FLAG affinity column. After extensive washing, the bound proteins were eluted with excess FLAG peptides, resolved and visualized by silver staining on SDS–PAGE. The protein bands on the gel were recovered and analysed by mass spectrometry. Representative peptide fragments of CEP131 and peptide coverage of the indicated proteins are shown. Detailed results from the mass spectrometric analysis are provided as Supplementary Data 1. (**b**) Whole-cell lysates from HeLa, MCF-7, HEK293T and U2OS cells were immunoprecipitated (IP) followed by immunoblotting (IB) with antibodies against the indicated proteins. (**c**) Immunoaffinity purification of CEP131-containing protein complexes. Representative peptide fragments of USP9X and peptide coverage of the indicated proteins are shown. Detailed results are provided as Supplementary Data 1. (**d**) Whole cellular extracts from HeLa cells were fractionated on Superose 6 size exclusion columns. Chromatographic elution profiles and IB analysis of the chromatographic fractions with antibodies against the indicated proteins are shown. The elution positions of calibration proteins with known molecular masses are indicated, and an equal volume from each fraction was analysed. (**e**) Co-immunoprecipitation analysis of the molecular interface required for the interaction between CEP131 and USP9X. The conserved domains of CEP131 and USP9X were determined by the SMART programme. (**f**) Pull-down analysis of the domains involved in the interaction between USP9X and CEP131 with His-tagged USP9X deletion mutants purified from Sf9 cells and GST-tagged full length of CEP131 purified from bacteria cells. (**g**) Pull-down analysis of the domains involved in the interaction between CEP131 and USP9X with GST-tagged deletion mutants of CEP131 purified from bacteria cells and His-tagged USP9X/M purified from Sf9 cells.

protein CEP131 suggests that USP9X is a centrosome-associated protein. Immunofluorescent and microscopy analysis indicated that USP9X was co-localized with the centrosomal markers Centrin and γ-tubulin (Fig. 2a). Strikingly, the physical association of USP9X with centrosomes appeared to be dynamic and cell cycle-dependent: the centrosomal localization of USP9X was evident in S and $G_2$ phases, weakly detected in $G_1$ phase and largely diminished in metaphase (Fig. 2b and Supplementary Fig. 2a). Similar observations were also obtained in normal human mammary epithelial cells (HMECs; Supplementary Fig. 2b).

The dynamic pattern of cellular distribution of centrosomal USP9X is reminiscent of that of CEP131 during cell cycle reported previously[13]. Indeed, we demonstrated that USP9X is co-localized with CEP131 at centrosome (Fig. 2c). Moreover, immunofluorescent staining also confirmed the co-localization of USP9X with PCM1 (Fig. 2c), a key factor that restricts CEP131 in centriolar satellite[13]. Furthermore, knockdown of the expression of CEP131 resulted in a diminished centrosomal localization of USP9X in U2OS cells (Fig. 2d), suggesting that the centrosomal localization of USP9X is dependent on CEP131.

Next, we investigated the abundance of USP9X and CEP131 during cell cycle and showed that the protein levels of both USP9X and CEP131 oscillated at a similar pace during cell cycle progression: both were low in $G_1$ phase and elevated in S and $G_2$/M phases except 2 h post synchronization, a time when the abundance of CEP131, but USP9X, increased (Fig. 2e and Supplementary Fig. 2c,d). Consistently, co-immunoprecipitation analysis in synchronized cells demonstrated that the physical association between USP9X and CEP131 was detected primarily in S and $G_2$ phases of the cell cycle (Fig. 2f). Together, these results support the argument that USP9X is recruited by and co-localized with CEP131 to centrosome, suggesting that USP9X is yet another component of centrosome.

**USP9X is functionally linked to the stability of CEP131**. To address the functional significance of the physical interaction and spatial co-localization between USP9X and CEP131, we examined the effect of USP9X on the expression of CEP131. Western blotting (WB) analysis revealed that, while the expression of centrosome marker protein γ-tubulin and loading control protein β-actin was essentially unchanged, the level of CEP131 was markedly decreased upon USP9X depletion by distinct short interfering RNAs (siRNAs) in MCF-7 cells (Fig. 3a, left panel). Similar observation was obtained in U2OS cells (Supplementary Fig. 3a, left panel) and HMECs (Supplementary Fig. 3b, left panel), whereas knockdown of USP33 had no evident effect on the expression of CEP131 (Fig. 3b). In addition, the mRNA expression level of CEP131 was not affected upon USP9X depletion (Fig. 3a, right panel; Supplementary Fig. 3a, right panel and Supplementary Fig. 3b, right panel). Moreover, the reduction in CEP131 protein level associated with USP9X depletion was probably a result of proteasome-mediated protein degradation, as the effect could be effectively blocked by a proteasome-specific inhibitor, MG132 (Fig. 3c). These observations indicate that CEP131 is a substrate of UPS9X.

In support of this deduction, cycloheximide (CHX) chase assays revealed that USP9X depletion was associated with a decreased CEP131 half-life in MCF-7 cells (Fig. 3d and Supplementary Fig. 3c) and U2OS cells (Supplementary Fig. 3d). Moreover, USP9X overexpression was able to restore the expression of CEP131 and prolong the half-life of CEP131 in USP9X-deficient cells (Fig. 3e). Consistently, depletion of USP9X resulted in lost CEP131 staining in centrosome (Fig. 3f), and examination of the protein expression levels of USP9X and

CEP131 in multiple cell lines showed a correlated pattern of the expression of these two proteins (Fig. 3g). Together, these results further support the notion that USP9X controls the stabilization of CEP131.

To exclude the possibility that USP9X depletion-associated loss of CEP131 from centrosome is indirectly affected by other centrosome components, we examined the localization and abundance of PCM1 after USP9X depletion. The results demonstrated that the centrosomal localization of PCM1 was mildly interrupted (Supplementary Fig. 4a), and the protein, but not mRNA, expression level of PCM1 decreased (Supplementary Fig. 4b), suggesting that PCM1 is a potential substrate of USP9X. However, we found that in USP9X-deficient cells, FLAG-tagged CEP131 could be effectively recruited to centrosome (Supplementary Fig. 4c), suggesting that mildly disrupted centrosomal localization of PCM1 associated with USP9X depletion has limited effect on CEP131 recruitment. In support of this argument, we revealed that severe loss of PCM1 indeed impaired CEP131 centrosomal localization as reported[13], while mild loss of PCM1 failed to do so (Supplementary Fig. 4d). The expression level of CEP131 was essentially not altered upon PCM1 knockdown (Supplementary Fig. 4d). Furthermore, we examined the expression level and localization of Pericentrin and CEP290, both of which are essential for the centrosomal restriction of PCM1 and CEP131 (ref. 13), and the results indicated that the localization and abundance of these proteins were unaffected upon USP9X knockdown (Supplementary Fig. 4a,b). Collectively, these results indicate that USP9X depletion-associated CEP131 loss from centrosome is not a consequence of a general loss of satellite proteins.

**USP9X deubiquitinates CEP131**. To gain molecular insights into the functional connection between USP9X and CEP131, we examined whether USP9X-promoted CEP131 stabilization is dependent on the enzymatic activity of USP9X. To this end, we created two stable MCF-7 cell lines with Dox-inducible expression of wild-type USP9X (USP9X/wt) and catalytically inactive mutant of USP9X (USP9X/C1566S)[31], respectively. WB analysis showed that, in cells expressing USP9X/wt, but not USP9X/C1566S, the protein level of CEP131 dramatically increased in a Dox dose-dependent manner (Fig. 4a, left panel), while the mRNA expression level of CEP131 was unchanged (Fig. 4a, right panel).

To consolidate these observations, we utilized the CRISPR/Cas9 system to knockout USP9X and demonstrated that, in USP9X null cells, CEP131 was downregulated and the downregulation of CEP131 could be reverted by forced expression of USP9X/wt, but overexpression of USP9X/C1566S could only moderately revert the CEP131 downregulation (Supplementary Fig. 5a, upper panel), while the mRNA expression level of CEP131 was essentially unchanged (Supplementary Fig. 5a, lower panel). Moreover, treatment of MCF-7 cells with WP1130, a deubiquitinase inhibitor reported to inhibit USP9X at low micromolar concentrations[33], resulted in a dose-dependent reduction in the protein, but not mRNA, level of CEP131 (Fig. 4b). Together, these results indicate that USP9X regulates the stability of CEP131 through its deubiquitinase activity.

Next, immunoprecipitation analysis showed that USP9X knockdown resulted in increased levels of ubiquitinated CEP131 species (Fig. 4c and Supplementary Fig. 5b). Furthermore, we demonstrated that the levels of ubiquitinated CEP131 species decreased in a Dox dose-dependent manner in HeLa cells with Dox-inducible expression of USP9X/wt (Fig. 4d), but not USP9X/C1566S (Supplementary Fig. 5c). To further identify the preference lysine residue of USP9X-promoted CEP131

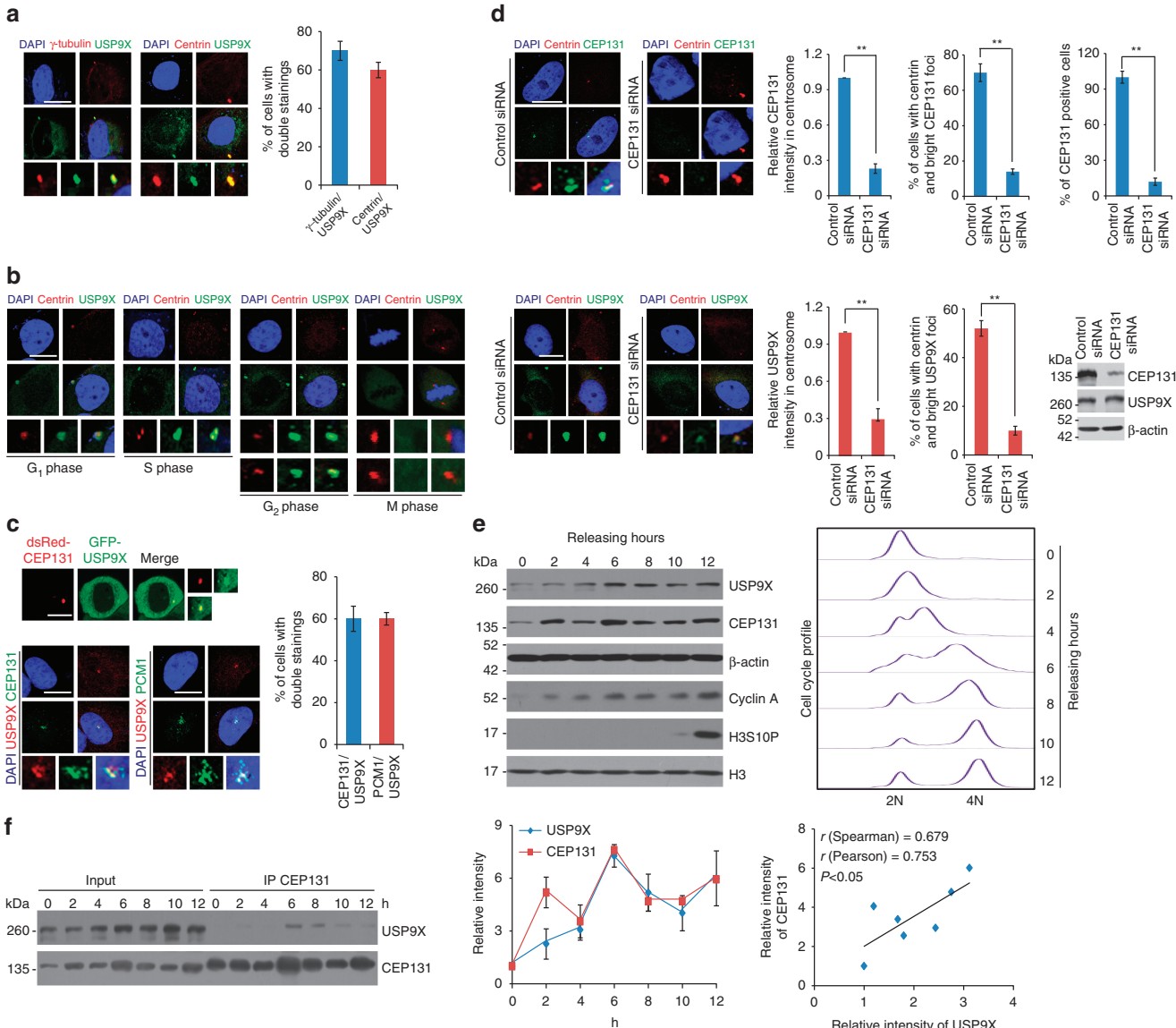

**Figure 2 | Co-localization of USP9X with CEP131 in centrosome.** (**a**) Confocal microscopy analysis of USP9X subcellular localization. The centrosome was stained with antibodies against γ-tubulin and Centrin. Percentage of cells with double staining was counted. Each bar represents the mean ± s.d. for biological triplicate experiments. Scale bar, 10 μm. (**b**) U2OS cells were synchronized by double-thymidine block and then released to G₁-, S-, G₂- or M phase followed by immunostaining with antibodies against Centrin and USP9X. Representative images from biological triplicate experiments are shown. Scale bar, 10 μm. (**c**) Co-localization of GFP-USP9X with dsRed-CEP131 in U2OS cells (upper panel). Immunostaining of U2OS cells with antibodies against USP9X and CEP131 or PCM1 followed by confocal microscopy analysis (lower left panel). Scale bars, 10 μm. Percentage of cells with double staining was counted (lower right panel). Each bar represents the mean ± s.d. for biological triplicate experiments. (**d**) U2OS cells transfected with control siRNA or CEP131 siRNA were fixed and subjected to immunostaining with antibodies against the indicated proteins. Scale bars, 10 μm. Percentage of cells with double staining or CEP131 knockdown was counted, and relative intensity of CEP131 or USP9X in centrosome was analysed by the Image J software. Each bar represents the mean ± s.d. for biological triplicate experiments. **P < 0.01, Student's t-test. The knockdown effect of CEP131 was examined by western blotting. (**e**) U2OS cells synchronized by double-thymidine block were released and cellular extracts were collected for western blotting analysis with antibodies against the indicated proteins. Representative cell cycle profiles are shown. Intensity of each band from biological triplicate experiments was quantified by densitometry with the Image J software with β-actin as a normalizer. Each bar represents the mean ± s.d. for biological triplicate experiments. The correlation coefficient and P values are shown. (**f**) U2OS cells synchronized by double-thymidine block were released and cellular extracts were collected for co-immunoprecipitation analysis of the association of CEP131 with USP9X.

deubiquitination, we utilized ubiquitin mutant with all lysine residues replaced by arginine except K29 (K29-only) or K48 (K48-only) or K63 (K63-only), or ubiquitin chain type-specific antibodies to differentiate ubiquitin species opposed by USP9X on poly-ubiquitination of CEP131. The results provided in Supplementary Fig. 5d,e indicated that K48-linked ubiquitin species are major forms opposed by USP9X. Moreover, *in vitro* deubi-

quitination assays with haemagglutinin (HA)-Ub-conjugated FLAG-CEP131 and FLAG-USP9X/wt or FLAG-USP9X/C1566S purified from HeLa cells revealed that USP9X/wt was capable of deubiquitinating CEP131, whereas USP9X/C1566S was not (Fig. 4e). Together, these results indicate that USP9X targets CEP131 for deubiquitination, supporting a notion that CEP131 is a *bona fide* substrate of USP9X.

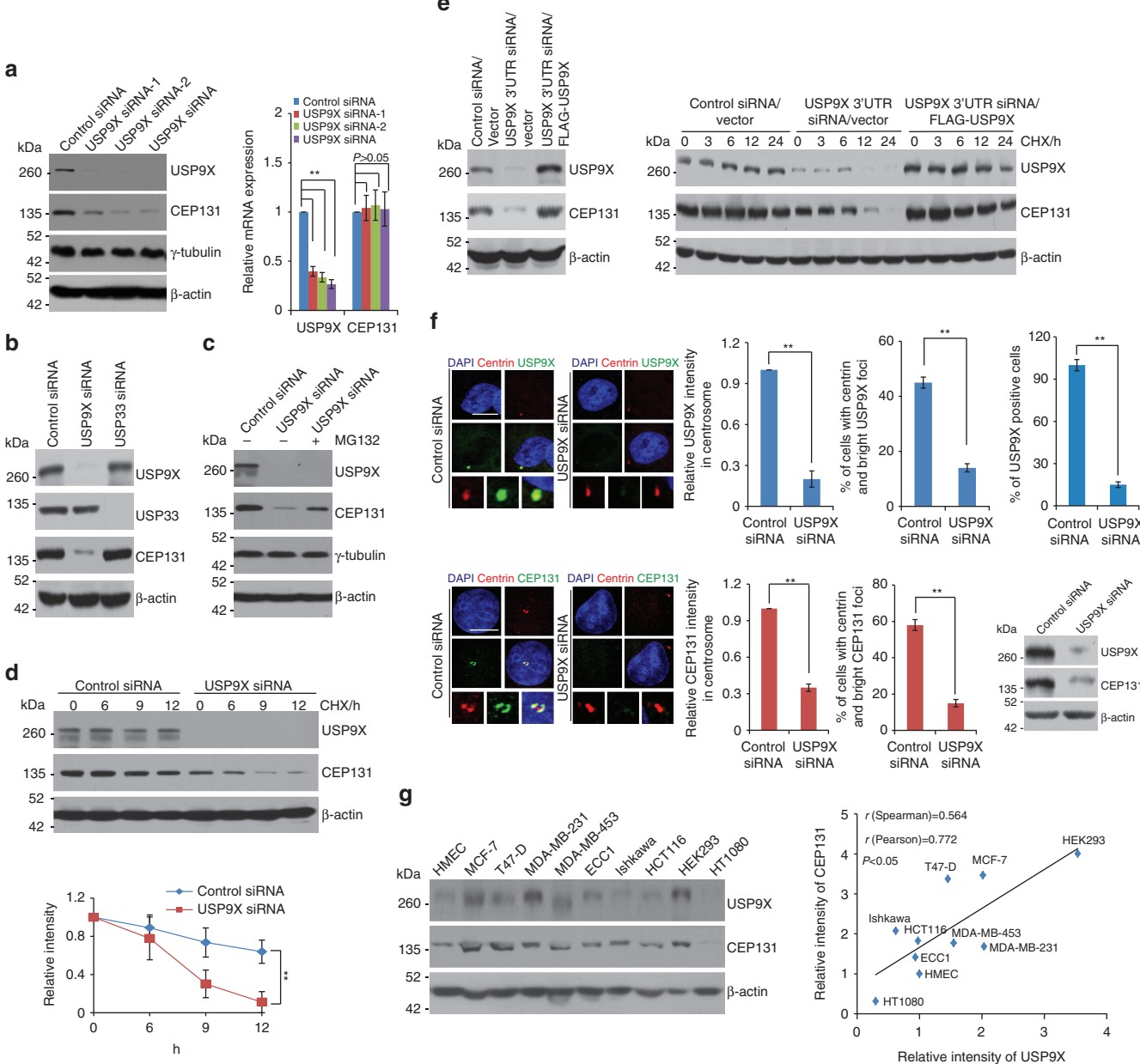

**Figure 3 | USP9X promotes CEP131 stabilization.** (**a**) MCF-7 cells were transfected with control siRNA or different sets of USP9X siRNAs. Cellular extracts and total RNA were prepared and analysed by western blotting and qRT (quantitative reverse transcription)–PCR, respectively. Each bar represents the mean ± s.d. for biological triplicate experiments. **$P < 0.01$, one-way analysis of variance (ANOVA). (**b**) MCF-7 cells were transfected with control siRNA, USP9X siRNA or USP33 siRNA followed by western blotting analysis. (**c**) MCF-7 cells were transfected with control siRNA or USP9X siRNA followed by treatment with dimethylsulphoxide or proteasome inhibitor MG132 (10 μM). Cellular extracts were prepared and analysed by western blotting. (**d**) MCF-7 cells transfected with control siRNA or USP9X siRNA were treated with CHX and harvested at the indicated time followed by western blotting analysis. Intensity of each band from biological triplicate experiments was quantified by densitometry with the Image J software with β-actin as a normalizer. Each bar represents the mean ± s.d. for biological triplicate experiments. **$P < 0.01$, two-way ANOVA. (**e**) Control U2OS cells or U2OS cells stably expressing USP9X were transfected with control siRNA or USP9X 3′UTR siRNA for 96 h followed by western blotting analysis (left panel). Control U2OS cells or U2OS cells stably expressing USP9X were transfected with control siRNA or USP9X 3′UTR siRNA for 96 h, then treated with CHX and harvested at the indicated time followed by western blotting analysis (right panel). (**f**) U2OS cells transfected with control siRNA or USP9X siRNA were fixed and subjected to immunostaining (left panel). Scale bars, 10 μm. Percentage of cells with double staining or USP9X knockdown was counted, and the relative intensity of USP9X or CEP131 in centrosome was analysed by the Image J software (right panel). Each bar represents the mean ± s.d. for biological triplicate experiments. **$P < 0.01$, two-tailed unpaired $t$-test. The knockdown effect of USP9X was examined by western blotting. (**g**) Western blotting analysis of the expression of USP9X and CEP131 in multiple cell lines. Intensity of each band was quantified by densitometry with the Image J software with β-actin as a normalizer. The correlation coefficient and $P$ values are shown.

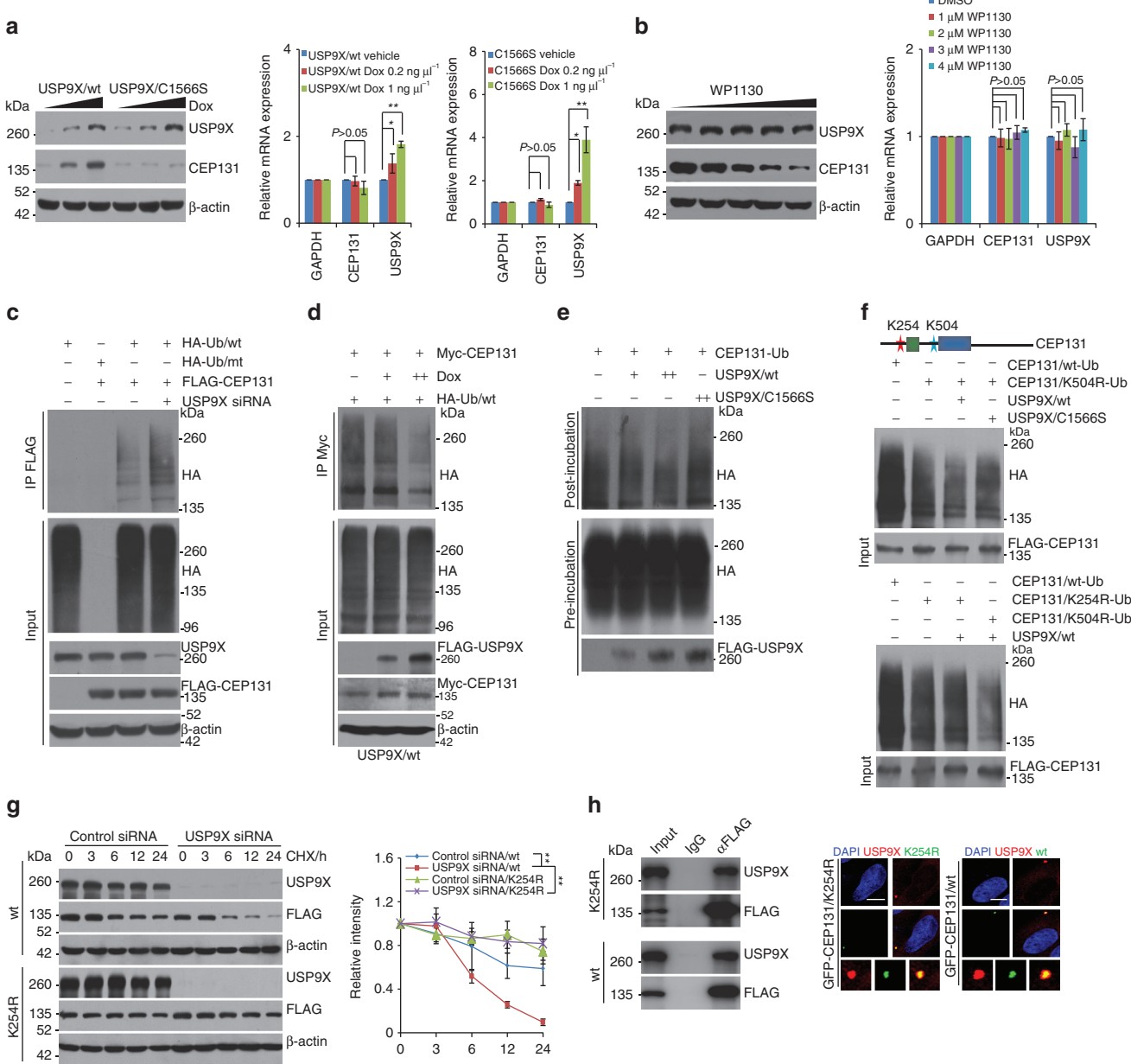

**Figure 4 | USP9X promotes CEP131 deubiquitination.** (**a**) MCF-7 cells with Dox-inducible expression of FLAG-USP9X/wt or FLAG-USP9X/C1566S were cultured in the absence or presence of increasing amounts of Dox. Cellular extracts and total RNA were collected and analysed by western blotting (left panel) and qRT–PCR (right panel), respectively. Each bar represents the mean ± s.d. for biological triplicate experiments. *$P < 0.05$, **$P < 0.01$, one-way ANOVA. (**b**) MCF-7 cells were cultured in the absence or presence of WP1130 as indicated. Cellular extracts and total RNA were collected and analysed by western blotting (left panel) and qRT–PCR (right panel), respectively. Each bar represents the mean ± s.d. for biological triplicate experiments. $P$ values were determined by one-way ANOVA. (**c**) MCF-7 cells stably expressing FLAG-CEP131 were co-transfected with control siRNA or USP9X siRNA together with HA-tagged wild type ubiquitin (Ub/wt) or a ubiquitin mutant (Ub/mt) with all lysine residues replaced by arginine as indicated. Cellular extracts were prepared for co-immunoprecipitation assays with anti-FLAG followed by IB with anti-HA. (**d**) HeLa cells with Dox-inducible expression of USP9X/wt were co-transfected with Myc-CEP131 and HA-Ub/wt and cultured in the absence or presence of Dox. Cellular extracts were prepared for co-immunoprecipitation assays with anti-Myc, followed by IB with anti-HA. (**e**) *In vitro* deubiquitination assays with HeLa cells-purified FLAG-tagged CEP131-Ub and USP9X/wt or USP9X/C1566S. (**f**) *In vitro* deubiquitination assays with HeLa cells-purified FLAG-tagged CEP131/K254R-Ub or CEP131/K504R-Ub and USP9X/wt or USP9X/C1566S. (**g**) MCF-7 cells stably expressing CEP131/wt or CEP131/K254R mutant were transfected with control siRNA or USP9X siRNA. Cells were treated with CHX and harvested at the indicated time followed by western blotting analysis. Intensity of each band from biological triplicate experiments was quantified by densitometry with the Image J software with β-actin as a normalizer. Each bar represents the mean ± s.d. for biological triplicate experiments. **$P < 0.01$, two-way ANOVA. (**h**) Co-immunoprecipitation (left panel) and microscopy analysis (right panel) of the association and co-localization of USP9X with CEP131/wt or CEP131/K254R. Representative images from biological triplicate experiments are shown. Scale bars, 10 μm.

SILAC-based quantitative analysis of human ubiquitin-modified proteome indicated that four lysine residues in CEP131, K254, K320, K504 and K510 were potential sites for ubiquitination, with K254 displaying the highest score and highest fold of change of the ubiquitination level upon proteasome inhibitor treatment[14]. Interestingly, this residue is located within the N-terminal region of CEP131 that interacts with USP9X (Fig. 1f,g). Consistently, mass spectrometry analysis of ubiquitin-conjugated CEP131 revealed two CEP131 peptides carrying ubiquitin-modified sites, one of them is at K254 (Supplementary Fig. 5f). To test whether K254 ubiquitination could be targeted by USP9X, we generated two CEP131 mutants, either lysine residue at 254 or 504 replaced by arginine (CEP131/K254R or CEP131/K504R). In vitro deubiquitination assays revealed that, while USP9X was able to remove ubiquitins of CEP131/K504R, but not that of CEP131/K254R (Fig. 4f), favouring the argument that USP9X targets K254 of CEP131 for deubiquitination, although polyubiquitin chains conjugated onto CEP131/K254R and CEP131/K504R were both dramatically reduced (Fig. 4f). Moreover, CHX chase assays showed that depletion of USP9X was associated with a decrease in the half-life of CEP131/wt, but not that of CEP131/K254R (Fig. 4g and Supplementary Fig. 5g), although K254R mutant neither disrupted the interaction of CEP131 with USP9X (Fig. 4h, left panel) nor affected the cellular distribution of CEP131 (Fig. 4h, right panel).

**USP9X regulates centrosome biogenesis and mitotic fidelity.** To understand the biological significance of USP9X-mediated deubiquitination/stabilization of CEP131, we next investigated the effect of USP9X on centrosome biogenesis. To this end, U2OS cells stably expressing USP9X/wt or USP9X/C1566S were synchronized by double-thymidine block. At 6 and 12 h after release from the $G_1$/S block when most cells were in S/$G_2$ and early M phases, respectively, centrosome duplication was analysed by immunofluorescent staining of γ-tubulin and Centrin. Notably, overexpression of USP9X/wt, but not USP9X/C1566S, was associated with a significant increase of centrosome amplification, manifested by the acquisition of more than four Centrin and two γ-tubulin foci (Fig. 5a), while neither USP9X/wt nor USP9X/C1566S affected cell cycle profile (Supplementary Fig. 6a). Remarkably, the phenotype evoked by USP9X over-expression could be largely rescued by CEP131 silencing (Fig. 5b). Similar observations were also obtained in MCF-7 cells (Supplementary Fig. 6b). These experiments suggest that USP9X promotes the generation of excessive centriolar foci through stabilizing CEP131.

We then investigated the effect of USP9X depletion on centrosome duplication. To this end, USP9X- or CEP131-depleted cells were treated with hydroxyurea (HU) to induce a prolonged S phase. Immunofluorescent and microscopy analysis showed that, while control cells arrested in S phase underwent centrosome overduplication, either USP9X or CEP131 depletion resulted in a significant reduction of centrosome amplification (Fig. 5c and Supplementary Fig. 6c), although mild defects in centrosome duplication were also observed in USP9X- or CEP131-depleted U2OS cells in the absence of HU treatment (Fig. 5c). In addition, the phenotype associated with USP9X deficiency could be rescued to certain extent by CEP131 overexpression (Fig. 5c and Supplementary Fig. 6c) or reverted by USP9X overexpression (Supplementary Fig. 6d). Meanwhile, in the absence or presence of HU, cell cycle profiles of these cells were essentially unchanged (Fig. 5c). Collectively, these results indicate that USP9X and CEP131 are important players in the regulation of centrosome duplication/biogenesis.

We next asked the question how USP9X-promoted CEP131 stabilization has an impact on centrosome biogenesis. A recent study reported that CEP131 is involved in centrosome duplication through regulating centrosomal localization of CDK2 (ref. 37), a cyclin-dependent kinase with established roles in centrosome biogenesis[38,39]. Indeed, similar to CEP131 depletion, USP9X deficiency was associated with an impaired centrosomal localization of CDK2 (Supplementary Fig. 6e), while the protein level of CDK2 was unaltered in both USP9X and CEP131 knockdown cells (Supplementary Fig. 6e). Importantly, USP9X depletion-induced phenotype of CDK2 dislocation could be rescued by CEP131 overexpression (Supplementary Fig. 6f). Moreover, USP9X-promoted centrosome amplification was abrogated upon CDK2 deletion (Supplementary Fig. 6g). These results point to a role of CEP131-regulated CDK2 localization in USP9X-promoted centrosome biogenesis.

Then, we examined whether USP9X-induced centrosome overduplication could result in chromosome instability and mitotic aberrations. Fluorescent microscopy and live cell imaging analysis showed that overexpression of USP9X or CEP131 was associated with an increase of mitotic aberrations characterized by a multipolar spindle or asymmetric, bipolar spindle with lagging chromosomes (Fig. 5d, Supplementary Fig. 7 and Supplementary Movie 1) and the phenotype associated with USP9X overexpression could be, at least partially, rescued by CEP131 silencing (Fig. 5d, Supplementary Fig. 7 and Supplementary Movie 1). Together, these results indicate that overexpression of USP9X promotes centrosome amplification and mitotic defects in a CEP131-dependent manner.

**USP9X and CEP131 are overexpressed in breast carcinomas.** Since centrosome amplification is a common feature of tumour cells[40–43], it would be interesting to know the expression level of USP9X/CEP131 in clinical carcinomas. Analysis of the integrated cancer DNA microarray database Oncomine (https://www.oncomine.com)[44] revealed that, compared to its expression in normal mammary tissues, USP9X is indeed markedly upregulated in breast carcinoma samples (Fig. 6a). Consistently, the expression level of USP9X was significantly higher in breast carcinoma samples than that in normal mammary tissues (Fig. 6b,c, left panel). The same is true in cultured breast cancer cells and normal breast epithelial cells (Fig. 3g). In agreement with our main argument, when the expression level of USP9X was plotted against that of CEP131, a strong positive correlation was found (Fig. 6c, right panel). Moreover, immunohistological analysis of the protein levels of USP9X and CEP131 using human tissue arrays that included breast carcinoma samples and normal mammary tissues showed that both USP9X and CEP131 were highly expressed in breast carcinoma samples (Fig. 6d, upper panel), and the levels of their expression were strongly correlated with each other (Fig. 6d, lower panel) and with the progression of breast cancer (Fig. 6d, upper panel). Collectively, these observations suggest a potential role for USP9X-promoted CEP131 stabilization in breast carcinogenesis.

As the protein abundance of PCM1 is also subjected to USP9X regulation (Supplementary Fig. 4b), we next examined the expression level of PCM1 in clinical samples. As demonstrated in Supplementary Fig. 8a,b, the protein abundance of PCM1 was indeed elevated in breast cancer and correlated with that of USP9X. However, similar results were not observed for CEP290, another essential satellite protein (Supplementary Fig. 8a,b), arguing against the possibility that the elevated expression of USP9X, CEP131 and PCM1 is resulting from aberrant centriolar satellites amplification.

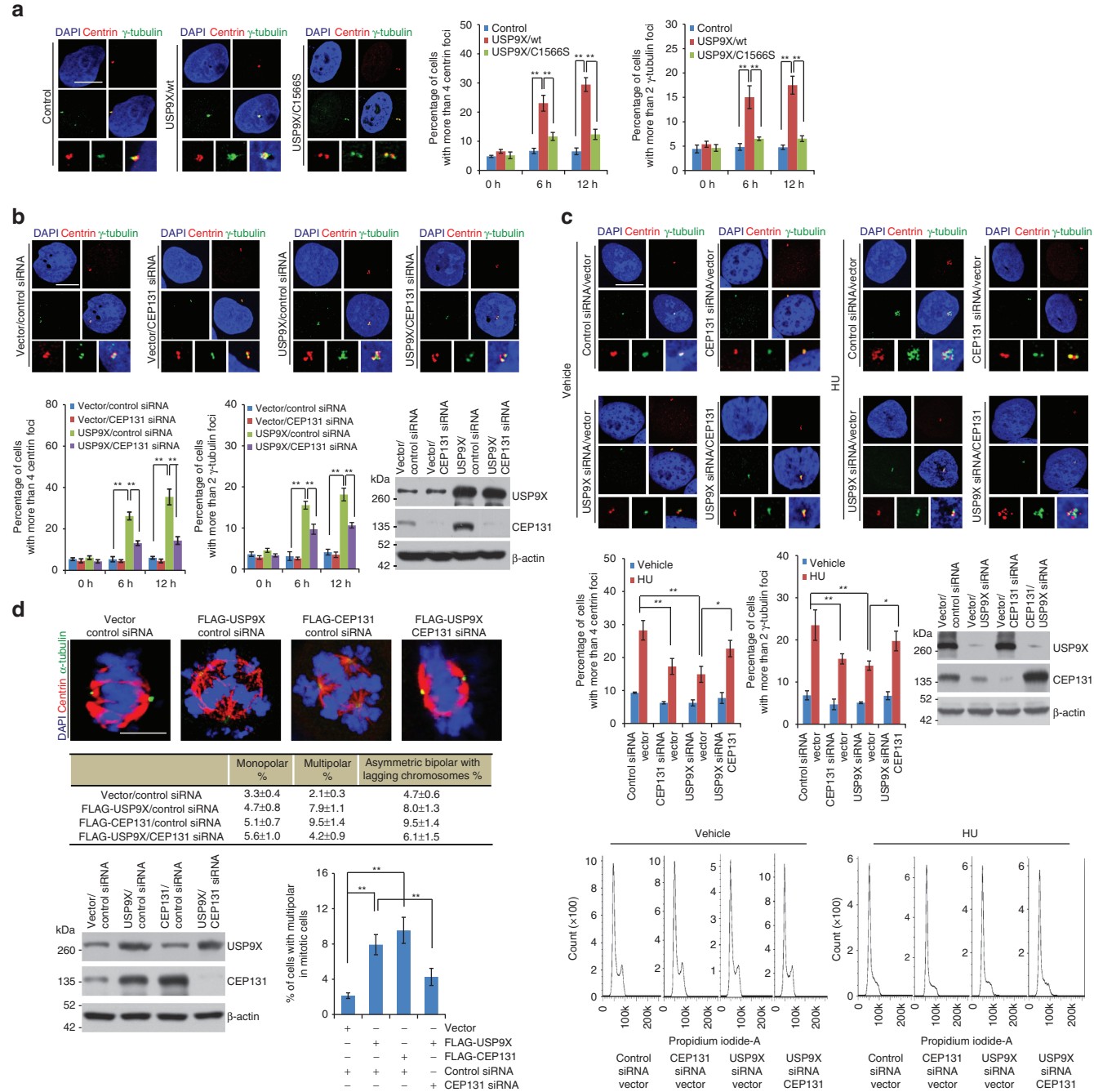

**Figure 5 | USP9X-promoted CEP131 stabilization regulates centrosome amplification and chromosome stability. (a)** Control U2OS cells and cells with Dox-inducible expression of USP9X/wt or USP9X/C1566S in the presence of Dox were synchronized with double-thymidine block and released followed by immunostaining with antibodies against the indicated proteins (left panel). Scale bar, 10 μm. Population of cells with the indicated numbers of foci at different time points were counted (right panel). Each bar represents the mean ± s.d. for biological triplicate experiments. **$P < 0.01$, one-way ANOVA. **(b)** U2OS cells stably expressing USP9X were transfected with indicated siRNAs. The cells were synchronized with double-thymidine block and released followed by immunostaining with antibodies against the indicated proteins (upper panel). Scale bar, 10 μm. Population of cells with the indicated numbers of foci at different time points were counted (lower left panel). Each bar represents the mean ± s.d. for biological triplicate experiments. **$P < 0.01$, one-way ANOVA. The expression of indicated proteins was examined by western blotting (lower right panel). **(c)** U2OS cells transfected with indicated siRNAs or genes were treated with HU, followed by analysis of centrosome numbers with immunostaining (upper panel). Scale bar, 10 μm. Population of cells with the indicated numbers of foci were counted. Each bar represents the mean ± s.d. for biological triplicate experiments (middle left panel). *$P < 0.05$; **$P < 0.01$, one-way ANOVA. The expression of USP9X and CEP131 from cells with HU treatment was examined by western blotting (middle right panel). Cell cycle profiles were determined by FACS (lower panel). **(d)** MCF-7 Cells stably expressing USP9X or CEP131 were transfected with indicated siRNAs and synchronized with double-thymidine block. Cells were collected after 12 h of release and stained (upper panel). Scale bar, 10 μm. Percentage of cells with different chromosomal instability phenotypes in mitosis was calculated and shown (middle panel). Percentage of cells with multipolar mitotic phenotype was analysed (lower panel). Each bar represents the mean ± s.d. for biological triplicate experiments. **$P < 0.01$, one-way ANOVA. The expression of indicated proteins was examined by western blotting (lower panel).

**USP9X/CEP131 axis promotes breast carcinogenesis.** In order to support a role of USP9X/CEP131 in breast carcinogenesis, we developed MCF-7 cells with Dox-inducible expression of stably integrated FLAG-USP9X or FLAG-CEP131. Colony formation assays showed that overexpression of USP9X or CEP131 was associated with moderate increases in colony numbers of MCF-7

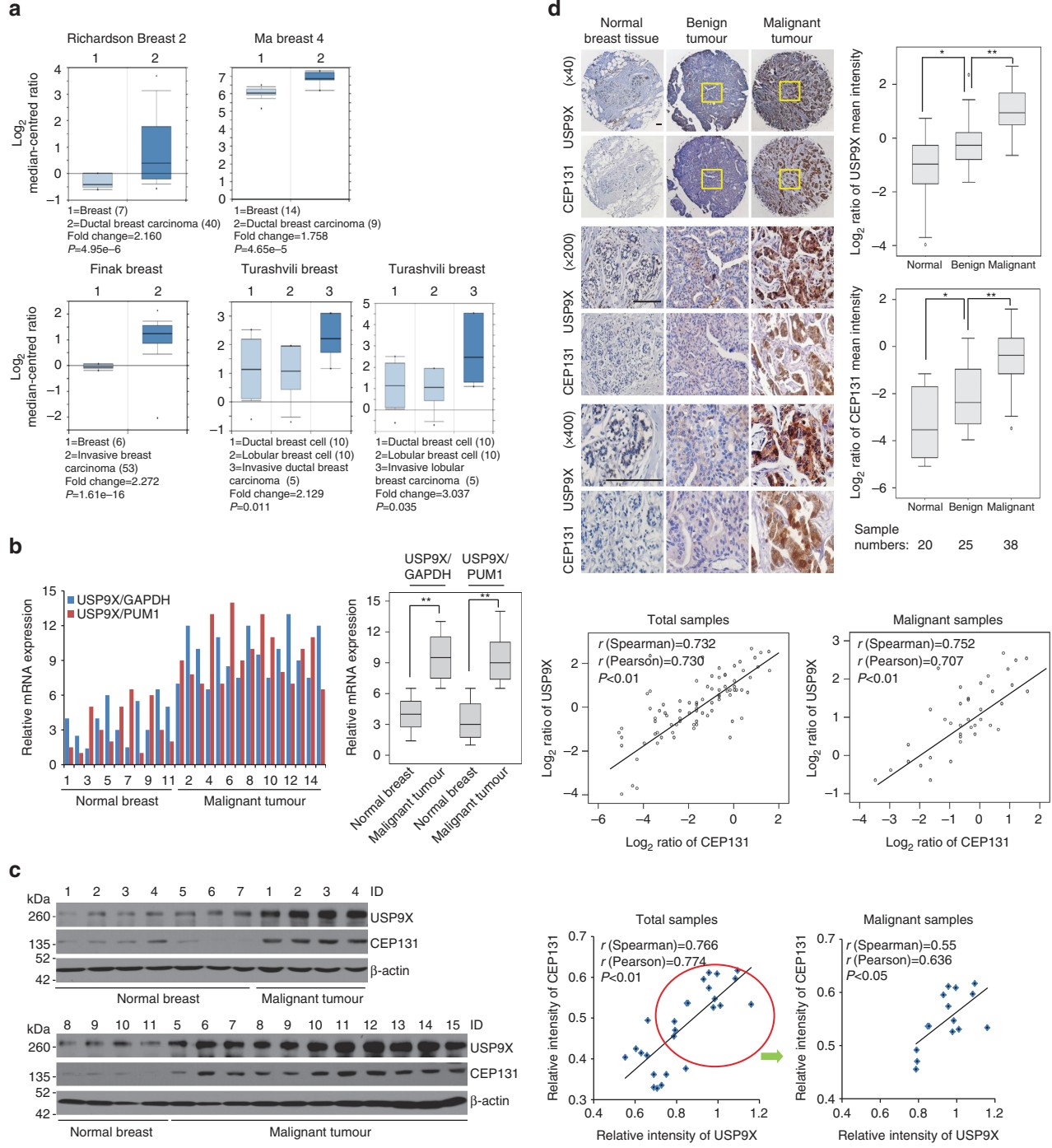

**Figure 6 | The expression of USP9X and CEP131 is elevated and correlated in breast cancer.** (**a**) Box plots of USP9X transcript levels in normal human mammary tissues and breast carcinoma samples based on four independent data sets from Oncomine. (**b**) The expression level of USP9X in 11 normal mammary tissues and 15 breast carcinomas was quantified by qRT–PCR and normalized as indicated. **$P < 0.01$, Student's $t$-test. (**c**) Cellular extracts from 11 normal mammary tissues and 15 breast cancer samples were prepared for western blotting analysis (left panel). The expression level of USP9X was quantified by the Image J software and normalized to β-actin. The correlation coefficient and $P$ values are shown (right panel). (**d**) Immunohistochemistry analysis of the expression levels of USP9X and CEP131 in breast tumours (upper left panel). Representative images were collected in three different magnification fields. Scale bars, 50 μm. The scores of the stained sections were determined by evaluating the intensity of immunopositivity by the Image-pro Plus software and were presented with box plots (upper right panel). Values that are less than or equal to the first quartile minus 1.5 times the interquartile range, or are greater than or equal to the third quartile plus 1.5 times the interquartile range are defined as outlier ones and indicated with a circle. *$P < 0.05$; **$P < 0.01$, one-way ANOVA. The correlation coefficient and $P$ values are analysed as indicated (lower panel).

cells (Fig. 7a, upper panel). Consistently, knockdown of USP9X or CEP131 severely impeded the colony formation of MCF-7 cells (Fig. 7a, lower panel). In addition, both USP9X-deficient and CEP131-deficient MCF-7 cells exhibited a much slower growth rate (Fig. 7b). Significantly, overexpression of CEP131 could, at least partially, offset the effect of USP9X knockdown on cell viability (Fig. 7c). Since centrosome dysregulation-associated mitotic defects could result in genome instability and cell apoptosis[13,45,46], we examined whether USP9X-promoted CEP131 stabilization plays a role in genome stability and cell death. Indeed, we demonstrated that either USP9X or CEP131 depletion resulted in marked accumulation of γH2AX (Supplementary Fig. 9a) and apoptosis of MCF-7 cells (Supplementary Fig. 9b). Moreover, USP9X depletion-associated effects could be alleviated by CEP131 overexpression (Supplementary Fig. 9a,b). These results indicate that USP9X-promoted CEP131 stabilization is required for breast cancer cell survival.

To investigate whether and how other substrates of USP9X might affect the cellular function of USP9X-promoted CEP131 stabilization, we analysed by WB analysis the expression of IPO5, PRMT5 and PPM1B, which were also identified as interactors of USP9X (Supplementary Fig. 1a), and the results indicate that the protein abundance of these proteins was essentially unchanged upon USP9X knockdown (Supplementary Fig. 10a). In agreement with previous reports[31,32,34,47], we did find that USP9X knockdown resulted in decreased expression of ITCH (Supplementary Fig. 10a), an E3 ligase involved in carcinogenesis[32,48], and MCL1, an anti-apoptotic regulator implicated in cancer[49,50]. However, CEP131 depletion had minimal effect on the expression level of ITCH and MCL1 (Supplementary Fig. 10a). In addition, USP9X, but not CEP131, could be co-immunoprecipitated by ITCH or by MCL1 (Supplementary Fig. 10b), and colony formation assay demonstrated that, although overexpression of ITCH or MCL1 could rescue the growth inhibitory phenotype associated with USP9X depletion to certain extent as CEP131 did, simultaneous expression of CEP131 and ITCH or MCL1 had an additive effect (Supplementary Fig. 10c). Moreover, USP9X depletion-associated defects of centrosome amplification could not be reverted by forced expression of either ITCH or MCL1 (Supplementary Fig. 10d), and knockdown of ITCH or MCL1 had no effect on centrosomal localization of USP9X and CEP131 (Supplementary Fig. 10e). These results suggest that CEP131 functions cooperatively with but independently of other USP9X substrates in USP9X-promoted breast cancer cell survival, and also provide an explanation for why overexpression of CEP131 could not fully restore the growth of USP9X knocked down tumour cells.

To further establish the role of USP9X/CEP131 in breast carcinogenesis, we transplanted USP9X- or CEP131-deficient MCF-7 cells onto the mammary fat pads of athymic mice. Notably, tumour growth in athymic mice receiving tumour transplants with depletion of either USP9X or CEP131 was greatly suppressed (Fig. 7d). Moreover, we demonstrated that overexpression of CEP131 in USP9X-deficient tumours could restore the growth of breast tumours (Fig. 7e). Next, we examined the centrosome numbers in cultured xenografts with immunofluorescent assays. The results in Fig. 7f showed that the percentage of cells with centrosome amplification was reduced in USP9X knocked down tumours, and this effect could be overridden by forced expression of CEP131. Together, these results support a notion that USP9X promotes breast carcinogenesis through stabilizing CEP131.

## Discussion

In this study, we found that the X-linked deubiquitinase USP9X is associated with centriolar satellite protein CEP131.

Immunofluorescent microscopy analysis revealed that in $G_1/S/G_2$ phases of the cell cycle USP9X is distributed not only in the cytoplasm but also in the centrosome, where it is physically and functionally associated with CEP131, indicating that USP9X is an integral component of the centrosome. Thus far, a number of studies describe USP9X as a cytoplasmic and membrane-associated protein[19,20], although mitochondrial[31] and nuclear localizations[51] of USP9X have also been reported. Our observation that USP9X is localized in the centrosome expands the spatial thus functional domains of this deubiquitinase. The diversified cellular localization of USP9X could be a reflection of a dynamic nature of distribution of this protein in cells. Indeed, it was reported that expressing cadherin cell adhesion molecules in fibroblasts altered USP9X localization[19]. We showed that the physical association of USP9X with centrosomes is cell cycle-dependent and mostly detected in S and $G_2$ phases. Remarkably, disrupting protein trafficking or the Golgi in polarized epithelia results in relocation or accumulation of USP9X, suggesting that this protein shuttles between a number of organelles and vesicles[52]. These observations highlight the need to interrogate USP9X–substrate interactions with approaches sorting specific subcellular compartments instead of more disruptive biochemical methods that extensively destroy cellular architectures[52].

A recently study in *Usp9x* knockout mouse identified disruption of the cytoskeleton as the main underlying consequence of the loss of *Usp9x* (ref. 22). Since cytoskeleton is composed of microtubules and actin filaments and centrosome is a microtubule-nucleation centre, our finding that USP9X is an integral component of centrosome provides a more appropriate explanation for *Usp9x* ablation-induced cytoskeleton collapse in mouse. We showed that USP9X, through stabilizing CEP131, regulates centrosome duplication and mitotic fidelity. In addition to our study, numerous reports showed the involvement of deubiquitination enzymes including BAP1, USP33, USP1, USP44, CYLD and USP21 in centrosome regulation and chromosomal stability[36,53–57], supporting a notion that deubiquitinases are the key regulators in centrosome homeostasis and genome stability.

CEP131 is a centriolar satellite protein and plays critical roles in the maintenance of genome stability[13,58]. Thus, understanding the mechanism by which the abundance of CEP131 is regulated is of great importance. It was reported that the expression of CEP131 could be induced by DNA methyltransferase inhibitor 5-azacytidine[13], and CEP131 is transcriptionally regulated by transcription factor SP1 (ref. 59). Our study indicates that USP9X, through its deubiquitinase activity, stabilizes CEP131 and regulates the abundance of CEP131 at a post-transcriptional level. Our observations support a model in which overexpression of USP9X in breast cancer results in elevated CEP131 protein, which, in turn, leads to centrosome amplification and genome instability, and eventually contributes to the development/progression of breast cancer.

Although we observed a better correlation between PCM1 and USP9X than that between CEP131 and USP9X in breast cancer samples, the effect of USP9X depletion on the expression of PCM1 was not as dramatic as that of CEP131, suggesting that PCM1 is a potential, but not major, substrate of USP9X in centrosome. These results likely explain why CEP131 overexpression could not fully compensate centrosomal biogenesis defects induced by USP9X depletion. Combining the findings that mildly disrupted centrosomal localization of PCM1 has minimal effect on CEP131 recruitment and the observations that USP9X directly interacts with CEP131 and opposes its polyubiquitin linkages *in vitro*, we get the conclusion that the effect of USP9X on CEP131 stabilization is attributed to the interplay between these two molecules but not through USP9X targeting other substrates like PCM1, and USP9X-regulated

PCM1 stabilization on centrosome activity, if it does so, seems to be independent of USP9X-promoted CEP131 stabilization. We believe that investigating whether/how PCM1 contributes to USP9X-regulated centrosome biogenesis will be helpful in understanding the functionality of USP9X in centrosome biology.

In addition to the canonical pathway, centrosome duplication can emerge from *de novo* assembly in which PCM clouds containing typical centrosomal proteins such as γ-tubulin and pericentrin provide localized environment supporting centriolar assembly[60]. Specifically, *de novo* centriole assembly can be

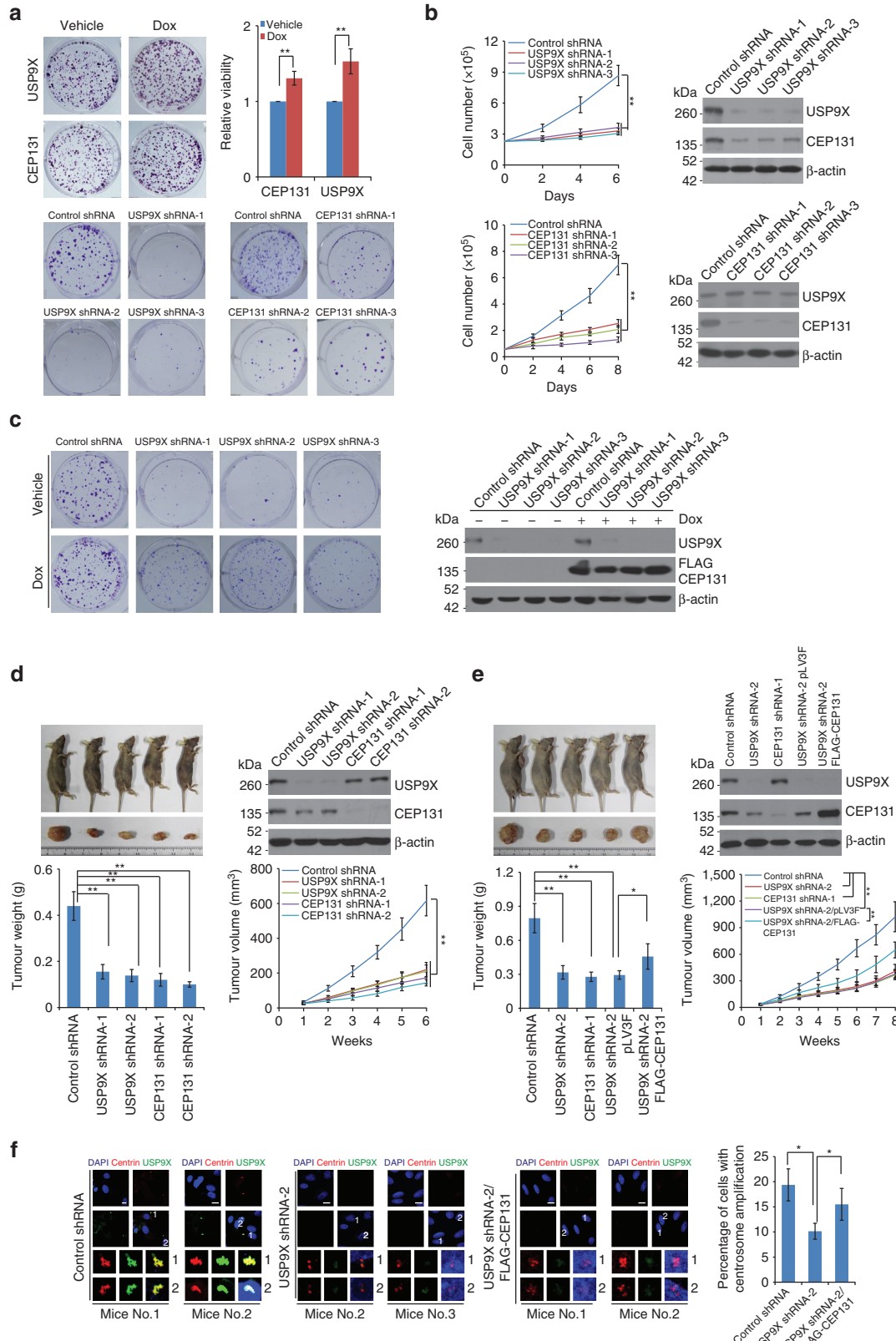

induced by overexpression of pericentrin[61]. A recent study indicates that ATF5, an essential PCM protein, interacts with both polyglutamylated tubulin on centriole and pericentrin in the PCM, and controls the centriole–PCM interaction[60]. It will be interesting to investigate whether CEP131 links PCM to centriole through ATF5 or/and CDK2, thereby regulating centrosome amplification and genome instability.

Among the polyubiquitin chains, K11, K29 and K48 poly-ubiquitin conjugates are considered as the most relevant ones associated with proteasome degradation[14,62–64], while K63 ubiquitin linkages are mainly involved in non-proteasomal pathways as a scaffolding modification in signal transduction[65]. Given that the physical association of USP9X with CEP131 was detected primarily in S and $G_2$ phases of the cell cycle, we did not investigate K11 ubiquitin linkages that are preferentially produced during mitosis and early $G_1$ (refs 62,66). In agreement with our observations that USP9X deubiquitinates K48 poly-ubiquitylated CEP131, USP9X has been reported to efficiently remove degradative K48-linked polyubiquitin chains on MCL1 (ref. 31) and XIAP[67]. Considering that the preference of USP9X on different types of ubiquitin linkages has been reported[31,67,68], we assume that USP9X opposes specific ubiquitin linkages in a substrate- or context-dependent manner.

A number of proteins involved in tumorigenesis have been reported to be substrates of USP9X (ref. 52). We believe that the link between CEP131 and USP9X is one of multiple pathways that appear to act in cancer cells. The observations that simultaneous expression of USP9X substrates rescued MCF-7 colony formation in an additive manner suggest that CEP131 functions cooperatively with but independently of other USP9X substrates and point to a role of USP9X at the apex of a regulation network that affects multiple cellular processes. Thereby, it will be interesting to explore the relationship between substrate diversity and cellular activities of USP9X. It will be also important to investigate the mechanisms underlying USP9X dysregulation in breast cancer and the role of the USP9X/CEP131 axis in the development/progression of breast cancer.

## Methods

**Antibodies and reagents.** The sources of antibodies against the following proteins were: CEP131 (sc-163722, 1:200 for immunofluorescence (IF)), WDR77 (sc-376556, 1:500 for WB), CEP290 (sc-70031, 1:200 for IF and 1:1,000 for WB), CDK2 (sc-6248, 1:200 for IF and 1:500 for WB), PCM1 (sc-398365, 1:200 for IF) and HA (sc-805, 1:500 for WB) from Santa Cruz Biotechnology; β-actin (A1978, 1:5,000 for WB), ITCH (SAB4200036, IP and 1:1,000 for WB), USP9X (WH0008239M1 for WB), γ-tubulin (T6557, 1:200 for IF) and FLAG (F3165, IP and 1:10,000 for WB) from Sigma; Histone H3S10 (05-1336, 1:2,000 for WB) and Centrin (04-1624, 1:500 for IF) from Millipore; α-tubulin (ab80779, 1:200 for IF), γ-tubulin (ab11317, 1:2,000 for WB), Pericentrin (ab28144, 1:200 for IF), CEP131

(ab99379, IP and 1:2,000 for WB), ubiquitin linkage-specific K48 (140601, 1:500 for WB), ubiquitin linkage-specific K63 (179434, 1:500 for WB) and Histone H3(ab1791, 1:10,000 for WB) from Abcam; USP9X (55054-1-AP, IP, 1:200 for IF and 1:2,000 for WB), CEP131 (25735-1-AP, IP, 1:200 for IF and 1:1,000 for WB), USP33 (20445-1-AP, 1:1,000 for WB), CP110 (12780-1-AP, IP and 1:1,000 for WB), PPM1B (13193-1-AP, IP and 1:1,000 for WB), SPTBN1 (19722-1-AP, 1:1,000 for WB), PFKFB3 (13763-1-AP, 1:1,000 for WB), PRMT5 (18436-1-AP, IP and 1:1,000 for WB), STK38 (11105-1-AP, 1:1,000 for WB), MCL1 (16225-1-AP, IP and 1:1,000 for WB) and PCM1 (19856-1-AP, 1:1,000 for WB) from Proteintech; IPO5 (MA1-10886, IP and 1:1,000 for WB) from Thermo Fisher Scientific and Myc (M047-3, IP and 1:1,000 for WB) from MBL. Anti-c-Myc agarose affinity gel (A7470), Myc peptide (M2435), anti-FLAG M2 affinity gel (A2220), 3 × FLAG peptide (F4799), anti-HA affinity gel (E6779), HU (H8627), thymidine (T1895), MG132 (SML1135) and Dox (D9891) were purchased from Sigma. CHX (0970, working concentration 50 μg ml$^{-1}$) was purchased from TOCRIS. Uncropped scans for blots from Figs 1–7 are shown in Supplementary Fig. 11 in the Supplementary Information.

**Plasmids.** The FLAG-tagged or Myc-tagged CEP131 carried by pLenti-Tight-Puro vector or pLenti-Hygro were amplified from CEP131 cDNA purchased from Open Biosystem. The dsRed-tagged CEP131 was created by integrating the dsRed cassette into pLenti-Hygro-CEP131 vector. The K254R and K504R CEP131 mutants were constructed by quick change strategy using a point mutation kit from Stratagene. The human FLAG-tagged USP9X and USP9X/C1566S mutant carried by pLenti-Tight-Puro vector or pLenti-Hygro vector were amplified from wild-type and catalytic mutant of V5-tagged USP9X cDNA kindly provided by Dr Feng Cong (Novartis Institutes for BioMedical Research, Cambridge), respectively. Green fluorescent protein (GFP)-tagged USP9X was created by incorporation of GFP open reading frame into the N-terminal of FLAG-USP9X-pLenti-Tight-Puro vector. Cyan fluorescent protein (CFP) was fused to the N terminal of cyclin A2 carried by pLenti-Neo vector. Serial truncations of FLAG-tagged CEP131 or USP9X were amplified from CEP131 cDNA or USP9X cDNA, respectively, and subcloned into pLenti-Hygro vector, whereas full length or truncations of GST fusion CEP131 were amplified and integrated into pGEX-4T-3 vector. Deletion mutants of USP9X expressed in insect cells were carried by pFastBac-HTA vector. The FLAG-tagged MCL1 or FLAG-tagged ITCH carried by pLenti-Neo vector was amplified from MCL1 cDNA or ITCH cDNA purchased from Origene or Open Biosystem, respectively. CRISPR/Cas9 constructs lentiCas9-Blast (Addgene plasmid #52962) and lentiGuide-Puro (Addgene plasmid #52963) were gifts from Dr Feng Zhang (Broad Institute, Cambridge). HA-tagged ubiquitin K48-only (Plasmid #17605) and K63-only (Plasmid #17606) were gifts from Dr Ted Dawson (Johns Hopkins University School of Medicine, Baltimore). HA-tagged ubiquitin K29-only was created by point mutational cloning from the HA-Ub/mt with all lysine residues replaced by arginine (gift from Dr Luyang Sun, Peking University Health Science Center).

**Cell culture.** MCF-7, U2OS, HeLa, HEK293, HEK293T, MDA-MB-231, MDA-MB-453, HCT116, T47-D, HT1080, HMEC and Sf9 cells were obtained from the American Type Culture Collection (Manassas, VA) and were cultured under the manufacturer's instructions. Ishkawa and ECC1 cells were kindly provided by Dr Myles Brown (Dana-Farber Cancer Institute, Boston) and were cultured in RPMI 1640 medium with 10% of fetal bovine serum (FBS, Biological Industries). Cells that allow protein expression under Dox treatment were created in two steps. First, cells were infected with lentivirus carrying rtTA and subjected to Neomycin selection. Subsequently, the established rtTA cells were infected with virus carrying pLenti-Tight-Puro vector that encodes USP9X or CEP131, followed by puromycin selection. All of the cells integrated with rtTA were cultured in

**Figure 7 | USP9X promotes breast carcinogenesis through regulating CEP131 abundance.** (**a**) Colony formation assays of MCF-7 cells with Dox-inducible expression of USP9X or CEP131 in the absence or presence of Dox (upper panel). **\*\*P < 0.01, Student's t-test. Colony formation assays of MCF-7 cells stably transfected with USP9X shRNA or CEP131 shRNA (lower panel). Representative images from biological triplicate experiments are shown. (**b**) MCF-7 cells stably transfected with USP9X shRNA or CEP131 shRNA were subjected to growth viability assay (left panel). Each bar represents the mean ± s.d. for biological triplicate experiments. **\*\*P < 0.01, two-way ANOVA. western blotting analysis of the indicated proteins is shown (right panel). (**c**) MCF-7 cells with Dox-inducible expression of CEP131 were stably transfected with USP9X shRNA for colony formation assays in the absence or presence of Dox. Representative images from biological triplicate experiments (left panel) and western blotting analysis of the indicated proteins (right panel) are shown. (**d**) MCF-7 infected with control lentiviruses or lentiviruses carrying shRNA against USP9X or CEP131 were transplanted on athymic mice (BALB/c; Charles River, Beijing, China) and tumour volumes were measured weekly. Each point represents the mean ± s.d. for different animal measurements (n = 6). **\*\*P < 0.01, one-way ANOVA for tumour weight analysis and two-way ANOVA for tumour volume analysis. The levels of indicated proteins in these tumours were examined by western blotting. (**e**) USP9X-deficient MCF-7 tumours stably transfected with control vector or vector encoding CEP131 were transplanted onto athymic mice and tumour volumes were measured weekly. Each point represents the mean ± s.d. for different animal measurements (n = 6). **\*P < 0.05; \*\*P < 0.01, one-way ANOVA for tumour weight analysis and two-way ANOVA for tumour volume analysis. The levels of indicated proteins in these tumours were examined by western blotting. (**f**) The xenograft tumours from four mice of each group as indicated were cultured and immunostained with antibodies against Centrin and USP9X. Scale bars, 10 μm. Representative images from two mice of each group are shown. More than one hundred cells from each xenograft were counted. Each bar represents the mean ± s.d. for measurements of four xenografts in each group. **\*P < 0.05, one-way ANOVA.

Tet Approved FBS and medium from Clontech. All of the cells were authenticated by examination of morphology and growth characteristics, and were confirmed to be mycoplasma-free.

**In vitro deubiquitination assay.** HeLa cells expressing full-length CEP131 and HA-ubiquitin were collected and then lysed in RIPA Buffer (300 mM NaCl, 0.5% sodium deoxycholate, 0.1% SDS, 1% Nonidet P-40 and 50 mM Tris-Cl, pH 8.0). The resulting lysate was purified with anti-FLAG affinity gel, eluted with 3 × FLAG peptide and then subjected to HA affinity gel to enrich HA-Ub-conjugated CEP131 (CEP131-Ub). HeLa cells expressing full-length USP9X (wild type or C1566S mutant) were collected and then lysed in RIPA Buffer (300 mM NaCl, 0.5% sodium deoxycholate, 0.1% SDS, 1% Nonidet P-40 and 50 mM Tris-Cl, pH 8.0). The resulting lysate was incubated with anti-FLAG affinity gel for 2 h and the beads were then washed five times with RIPA Buffer. Recombinant USP9X and CEP131-Ub were incubated in buffer containing 50 mM HEPES, pH 7.5, 10 mM 2-mercaptoethanol and 0.5 mM EDTA at 30 °C for 30 min. The reactions were stopped by boiling for 5 min in 5 × SDS–PAGE loading buffer followed by WB analysis with appropriate antibodies.

**Immunofluorescence.** Cells on glass coverslips (BD) were fixed with 2% paraformaldehyde and permeabilized with 0.2% Triton X-100 in PBS. Samples were then blocked in 5% donkey serum in the presence of 0.1% Triton X-100 and stained with the appropriate primary and secondary antibodies coupled to AlexaFluor 488 or 594 (Invitrogen). To avoid bleed-through effects in double-staining experiments, each dye was scanned independently in a multitracking mode. More than 90 cells in each treatment were scored in biological triplicate experiments.

**USP9X knockout cell generation.** USP9X knockout HMEC cells were generated by co-transfection of plasmid encoding FLAG-Cas9 (lentiCas9-Blast) and sgRNA plasmid (lentiGuide-Puro) targeting USP9X (USP9X sgRNA 5′-GTTGAT-CATGTCATCCAACT-3′). Forty-eight hours after transfection, cells were selected by blasticidin (5 μg ml$^{-1}$) and puromycin (1 μg ml$^{-1}$) for 2 days. Single colony was picked up for continuous culture and USP9X disruption was confirmed by WB analysis.

**Immunopurification and silver staining.** Lysates from HeLa cells stably expressing FLAG-USP9X or CEP131 were prepared by incubating the cells in lysis buffer containing protease inhibitor cocktail (Roche). Anti-FLAG immunoaffinity columns were prepared using anti-FLAG M2 affinity gel (Sigma) following the manufacturer's suggestions. Cell lysates were obtained from ∼5 × 10$^8$ cells and applied to an equilibrated FLAG column of 1 ml bed volume to allow for adsorption of the protein complex to the column resin. After binding, the column was washed with cold PBS plus 0.2% Nonidet P-40. FLAG peptide (Sigma) was applied to the column to elute the FLAG protein complex as described by the vendor. The elutes were collected and visualized on NuPAGE 4–12% Bis-Tris gel (Invitrogen), followed by silver staining with a silver staining kit (Pierce). The distinct protein bands were retrieved and analysed by liquid chromatography–tandem mass spectrometry (LC-MS/MS).

**Nano-HPLC-MS/MS analysis of USP9X or CEP131 protein complex.** To identify proteins associated with FLAG-USP9X or CEP131, LC-MS/MS analysis was performed using a Thermo Finnigan LTQ linear ion trap mass spectrometer in line with a Thermo Finnigan Surveyor MS Pump Plus HPLC system. Tryptic peptides generated were loaded onto a trap column (300SB-C18, 5 × 0.3 mm, 5 μm particle; Agilent Technologies, Santa Clara, CA), which was connected through a zero dead volume union to the self-packed analytical column (C18, 100 μm i.d × 100 mm, 3 μm particle; SunChrom, Germany). The peptides were then eluted over a gradient (0–45% B in 55 min, 45–100% B in 10 min, where B = 80% acetonitrile, 0.1% formic acid) at a flow rate of 500 nl min$^{-1}$ and introduced online into the linear ion trap mass spectrometer (Thermo Fisher Corporation, San Jose, CA) using nano electrospray ionization. Data-dependent scanning was incorporated to select the five most abundant ions (one microscan per spectra; precursor isolation width 1.0 $m/z$, 35% collision energy, 30 ms ion activation, exclusion duration: 90 s; repeat count: 1) from a full-scan mass spectrum for fragmentation by collision induced dissociation. MS data were analysed using SEQUEST (v. 28) against NCBI human protein database (14 December 2011 downloaded, 33,256 entries), and results were filtered, sorted and displayed using the Bioworks 3.2. Peptides (individual spectra) with Preliminary Score (Sp) ≥ 500; Rank of Sp (RSp) ≤ 5; and peptides with + 1, + 2 or + 3 charge states were accepted if they were fully enzymatic and had a cross-correlation (Xcorr) of 1.90, > 2.75 and > 3.50, respectively. The following residue modifications were allowed in the search: carbamidomethylation on cysteine as fix modification and oxidation on methionine as variable modification. Peptide sequences were searched using trypsin specificity, allowing a maximum of two missed cleavages. Sequest was searched with a peptide tolerance of 3.0 Da and a fragment ion tolerance of 1.0 Da.

**FPLC chromatography.** HeLa cell nuclear extracts or FLAG-CEP131-containing protein complexes were applied to a Superose 6 size exclusion column (GE Healthcare) that had been equilibrated with dithiothreitol-containing buffer and calibrated with protein standards (Amersham Biosciences). The column was eluted at a flow rate of 0.5 ml min$^{-1}$ and fractions were collected.

**LC-MS/MS analysis of CEP131 ubiquitination sites.** The tryptic peptides of CEP131-conjugated ubiquitin were dissolved in 0.1% formic acid (FA), directly loaded onto a reversed-phase pre-column (Acclaim PepMap 100, Thermo Scientific). Peptide separation was performed using a reversed-phase analytical column (Acclaim PepMap RSLC, Thermo Scientific). The gradient comprises an increase from 2 to 35% solvent B (0.1% FA in 98% acetonitrile (ACN)) over 12 min and climbing to 80% in 4 min and then holding at 80% for the last 4 min, all at a constant flow rate of 400 nl min$^{-1}$ on an EASY-nLC 1000 UPLC system. The resulting peptides were analysed by Q Exactive hybrid quadrupole-Orbitrap mass spectrometer (Thermo Fisher Scientific). The peptides were subjected to NSI source followed by MS/MS in Q Exactive (Thermo) coupled online to the UPLC. Intact peptides were detected in the Orbitrap at a resolution of 70,000. Peptides were selected for MS/MS using NCE setting as 28; ion fragments were detected in the Orbitrap at a resolution of 17,500. A data-dependent procedure that alternated between one MS scan followed by 20 MS/MS scans was applied for the top 20 precursor ions above a threshold ion count of 5E3 in the MS survey scan with 10.0 s dynamic exclusion. The electrospray voltage applied was 2.0 kV. Automatic gain control was used to prevent overfilling of the Orbitrap; 5E4 ions were accumulated for generation of MS/MS spectra. For MS scans, the $m/z$ scan range was 350–1,800. Fixed first mass was set at 100 $m/z$. The resulting MS/MS data were processed using the Mascot search engine (v.2.3.0). Tandem mass spectra were searched against CEP131 (Homo sapiens) database. Trypsin/P was specified as cleavage enzyme allowing up to four missing cleavages for CEP131. Mass error was set to 10 p.p.m. for precursor ions and 0.02 Da for fragment ions. Carbamidomethyl on Cys was specified as fixed modification and oxidation on Met; acetylation on Protein N-term were specified as variable modifications for CEP131. Specifically, ubiquitination on lysine was set as variable modification for CEP131. Peptide ion score was set to > 20. Finally, two ubiquitination sites were identified in sample CEP131 (the protein coverage is 61.22%). All the detailed information was presented in Supplementary Fig. 5e.

**Immunoprecipitation.** Cell lysates were prepared by incubating the cells in NETN buffer (50 mM Tris-HCl, pH 8.0, 150 mM NaCl, 0.2% Nonidet P-40, 2 mM EDTA) in the presence of protease inhibitor Cocktails (Roche) for 20 min at 4 °C. This was followed by centrifugation at 14,000g for 15 min at 4 °C. For immunoprecipitation, ∼500 μg of protein was incubated with control or specific antibodies (1–2 μg) for 12 h at 4 °C with constant rotation; 50 μl of 50% protein G magnetic beads (Invitrogen) was then added and the incubation was continued for an additional 2 h. Beads were then washed five times using the lysis buffer. Between washes, the beads were collected by magnetic stand (Invitrogen) at 4 °C. The precipitated proteins were eluted from the beads by re-suspending the beads in 2 × SDS–PAGE loading buffer and boiling for 5 min. The boiled immune complexes were subjected to SDS–PAGE, followed by immunoblotting with appropriate antibodies.

**RNA interference.** All siRNA transfections were performed using Lipofectamine RNAi MAX (Invitrogen) following the manufacturer's recommendations. The final concentration of the siRNA molecules is 10 nM and cells were collected 72 or 96 h later according to the purposes of the experiments. Control siRNA (ON-TARGETplus Non-Targeting Pool, D-001810-10), USP9X siRNA (ON-TARGETplus, L-006099-00-0005) and CEP131 siRNA (ON-TARGETplus, L-023335-00-0005) were from Dharmacon in a smart pool manner, while the individual siRNAs against USP9X (USP9X siRNA-1: 5′-GTCGTTACAGCTAGTATTT-3′, USP9X siRNA-2: 5′-CTGTGATTCAGCAACTCTA-3′, USP9X siRNA-3/3′UTR: 5′-GAGAGTTTATTCACTGTCTTA-3′), PCM1 (PCM1 siRNA-1: 5′-CCAAT-GATATTTCTCCGGA-3′, PCM1 siRNA-2: 5′-CAGACTTCCCTCCAGGCTA-3′), CDK2 (5′-GAGCUUAACCAUCCUAAUA-3′), MCL1 (5′-GAAATTCTTT-CACTTCATT-3′), ITCH (5′-ACATGCCATCTACCGTCATTA-3′) and USP33 (5′-GAUCAUGUGGCGAAGCAUA-3′) were chemically synthesized by Sigma (Shanghai, China). The short hairpin RNAs (shRNAs) against USP9X and CEP131 in pLKO vectors were purchased from Sigma.

**Real-time RT–PCR.** Total cellular RNAs were isolated with TRIzol reagent (Invitrogen) and used for first strand cDNA synthesis with the Reverse Transcription System (Roche). Quantitation of all gene transcripts was done by qPCR using a Power SYBR Green PCR Master Mix (Roche) and an ABI PRISM 7500 sequence detection system (Applied Biosystems) with the expression of GAPDH or PUM1 as the internal control.

**Sequences of shRNAs and primers.** Sequences of shRNAs and sequences of primers used in quantitative PCR are provided in Supplementary Tables 1 and 2, respectively.

**Lentiviral production.** The shRNAs targeting CEP131 and USP9X in the pLKO.5 vector or vectors encoding rtTA, USP9X and CEP131 carrying by pLenti-Neo, pLenti-Hygro or pLenti-Tight-Puro, as well as three assistant vectors, pMDLg/pRRE, pRSV-REV and pVSVG, were transiently transfected into HEK293T cells. Viral supernatants were collected 48 h later, clarified by filtration and concentrated by ultracentrifugation.

**Recombinant protein purification.** Recombinant baculovirus carrying deletion mutants of USP9X was generated with the Bac-to-Bac System (Invitrogen). Infected Sf9 cells were grown in spinner culture for 48–96 h at 27 °C and His-tagged proteins were purified using $Ni^{2+}$-NTA agarose (Invitrogen) according to the standard procedures. Full length or deletion mutants of CEP131 were purified from bacteria BL21 cells with Glutathione-agarose.

**In vivo deubiquitination assay.** Cells with different treatments were lysed in RIPA buffer containing 50 mM Tris-HCl (pH 7.4), 150 mM NaCl, 1% NP-40, 0.1% SDS and protease inhibitor at 4 °C for 30 min with rotation, and centrifuged at 20,000$g$ for 15 min. About 0.5–1.5 mg of cellular extracts were immunoprecipitated with anti-FLAG or anti-Myc agarose affinity gel for 2 h. The beads were then washed five times with RIPA buffer, boiled in SDS loading buffer and subjected to SDS–PAGE followed by immunoblotting.

**Cell flow cytometry.** Cells with different treatments were trypsinized, washed with PBS and fixed in 70% ethanol at 4 °C overnight. After being washed with PBS, cells were incubated with RNAase A (Sigma) in PBS for 30 min at 37 °C and then stained with 50 mg ml$^{-1}$ propidium iodide. Cell cycle data were collected with FACS Calibur (Becton Dickinson) and analysed with the FlowJo software. Apoptosis of cells was analysed with FACS using Cells Annexin V Apoptosis Detection Kits as per the manufacturer's standard procedures (Affymetrix eBioscience).

**Colony formation assay.** MCF-7 cells stably expressing USP9X or CEP131 and shRNAs targeting CEP131 or USP9X were maintained in culture media for 14 days, followed by staining with crystal violet.

**Tissue specimens.** The samples of carcinomas and the adjacent normal tissues were obtained from surgical specimens from patients with breast cancer. Samples were frozen in liquid nitrogen immediately after surgical removal and maintained at − 80 °C until mRNA and protein extraction. Human breast tissue arrays were prepared, incubated with antibodies against USP9X, or CEP131, and processed for immunohistochemistry with standard 3,3'-diaminobenzidine (DAB)-staining protocols. Representative images for normal (tumour-adjacent normal breast tissue 20), benign (hyperplasia of duct 7, adenosis 13 and fibroadenoma 5) and malignant (intraductal carcinoma 17, invasive ductal carcinoma 18 and invasive lobular carcinoma 3) breast tumour samples are collected in three different magnification fields. All studies were approved by the Ethics Committee of the Tianjin Medical University, and informed consent was obtained from all patients.

**Tumour xenografts.** MCF-7 cells were plated and infected in vitro with lentiviruses carrying control shRNA, USP9X shRNA or CEP131 shRNA together with or without FLAG-tagged CEP131 at MOI of 100. Forty-eight hours after infection, $8 \times 10^6$ viable MCF-7 cells in 200 μl PBS were injected into the mammary fat pads of 6- to 8-week-old athymic mice (BALB/c; Charles River, Beijing, China). Animals were randomly assigned into five groups (six mice per group). Sample size estimate was based on xenograft assays from literatures. 17-β-oestradiol (E2) pellets (0.72 mg per pellet, 60 day release; Innovative Research of America, Sarasota, FL) were implanted 1 day before the tumour cell injection. Tumours were measured weekly using a vernier calliper and the volume was calculated according to the formula: $\pi/6 \times length \times width^2$. The measurement and data processing were done with blinding. For immunofluorescent assays with xenograft tumours, the frozen tumours were cut into pieces followed by trypsinization. Then, the cultured tumour cells were further enriched by puromycin selection (puromycin-resistant gene together with shRNA cassette carried by the shRNA-expressing lentivirus has been integrated into the genome of tumour cells). All animals were killed at the end of the experiment and included into the analysis. The study was approved by the Animal Care Committee of Tianjin Medical University.

**Statistical analysis.** Data from biological triplicate experiments are presented with error bar as mean ± s.d. Two-tailed unpaired Student's t-test was used for comparing two groups of data. Analysis of variance in conjugation with Bonferroni's correction was used to compare multiple groups of data. All of the statistical testing results were determined by the SPSS 19.0 software. Before statistical analysis, variation within each group of data and the assumptions of the tests were checked.

**Data availability.** Datasets of "Richardson breast 2", "Ma breast 4", "Finak breast", and "Turashvili breast" from Oncomine database (https://www.oncomine.com) were used to analyse differential expression of USP9X in breast cancer and normal breast tissues. All other remaining data are available within the Article and its Supplementary Files, or available from the authors upon request.

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

## Acknowledgements

This work was supported by grants (81272284, 91219102 and 31671474 to L.S. and 91219201, 81530073 and 81130048 to Y.S.) from the National Natural Science Foundation of China, and a grant (2011CB504204 to Y.S.) from the Ministry of Science and Technology of China. This work was also supported by Talent Excellence Program from Tianjin Medical University.

## Author contributions

Xin L., N.S., Y.S. and Lei S. designed the research studies; Xin L., N.S., L.L. and Xinhua L. conducted experiments; Xin L., N.S., L.L., X.S., Xinhua L., S.M., Lin S., X.Z., Yue W., D.S., S.Y., C.C. and Q.Z. acquired data; Xin L., N.S., Xinhua L., X.D., N.Y., F.Y., Yan W., Z.Y., Y.S. and Lei S. analysed data; Xin L., N.S., Y.S. and Lei S. wrote the manuscript.

## Additional information

**Publisher's note**: 

