## [Peer Review File · Nature Communications]

Reviewers' comments:

Reviewer #1 (Remarks to the Author):

Li et al

This is a provocative study of the potential requirement of the deubiquitinase USP9X to regulate levels of a centriolar satellite protein, CEP131, and so contribute to the regulation of centrosome numbers and then correlate this with tumour development. The difficulty with the paper is that everything is analysed at a very shallow level and many interpretations of some of the data are possible. The authors need to focus upon one aspect of this story and really get to understand it before publication would be possible.

The first difficulty of the paper is that it attempts to give a role to USP9X in regulating CEP131 function when we do not understand what CEP131 - or indeed centriolar satellites - actually do. The authors set out to study USP9X partners and identify CEP131. Unfortunately they do not really indicate in full the other proteins that they identify as interactors and there must be many. Part of the difficulty appears to be that they present data for gel purified interacting proteins rather than presenting unbiased analysis of all interactors identified in the pull down. This is important particularly in interpreting the subsequent experiments in which USP9X function is removed. I am certain that they are correct and that CEP131 is one of USP9X's partners - they present good evidence for this in Fig. 1 - but what are ALL its other partners? Nevertheless, an association with centrosomes (Fig. 2) makes sense if USP9X is a CEP131 partner.

The effects shown in subsequent figures and used to reach conclusions about USP9X function are in many cases modest. For example, the loss of CEP131 following USP9X RNAi in MCF7 cells is not convincing (Fig. 3B). Moreover, the change in CEP131 half-life following USP9X RNAi particularly in U2OS cells is certainly not dramatic. Similarly, the increase in CEP131 following expression of elevated USP9X is not dramatic (Fig. 4A), as claimed, and neither are the diminution of CEP131 levels following inhibition of the DUB with WP1130 (Fig. 4D).

It seems quite likely that CEP131 is a USP9X substrate but the question is what are its other substrates and how might these affect, not only the centrosome cycle, but more broadly oncogenesis. Without this knowledge it is difficult to interpret the meanings of Figures 6 and 7 that largely analyse correlative findings.

In conclusion, much more needs to be done to understand the roles of CEP131 in centrosome biology and the full range of substrates of USP9X before it will be possible to correctly interpret this study.

Reviewer #2 (Remarks to the Author):

In the current submission Li et al. investigate the function of deubiquitination enzyme USP9X in chromosome instability. This manuscript shows that USP9X binds directly to CEP131 and rescuing the levels of CEP131 in cancer cells. Upregulated CEP131 and USP9X in turn promoted chromosome instability in cancer cells. Next, the authors demonstrate that USP9X is overexpressed in breast carcinomas with worse survivals of these patients. Furthermore, overexpression of USP9X in cancer cells transplanted into mice caused breast carcinogenesis *in vivo*. The experimental evidence presented in the study is interesting and novel. They clearly show that USP9X regulates CEP131 stability in breast cancer however, the reviewer is not convinced with the authors' claim/evidence that presented regarding the function USP9X in non-cancerous cells. In addition, it is not clear how chromosome instability presented in the first three figures can be coupled to increase survival and proliferation of cancer cells presented in the last figure. Including data using USP9X cells isolated from knockout animals could strength the conclusion of this manuscript.

Major questions:

- What is the function and role of USP9X in non-cancerous cells? If the answer is that the expression of USP9X is low in normal cells, this is not clear by the data shown in Fig. 3F.
- USP9X transgenic animal has been generated previously. Including experiments using USP9X primary cells isolated from knockout animals can strength the conclusion of the authors. This important since all of the experiments present in the manuscript are performed by overexpressing/siRNA treating human cancer cell lines.
- Fig. 1E is not necessary and can be presented in Suppl. figures since the same information is presented in presented in 1D.
- The authors need to confirm direct interaction between USP9X and CEP131 using purified proteins. Fig. 1G shows the binding between "truncated mutant" of USP9X and CEP131.
- Downregulation of USP9X leads to reduced levels of CEP131. What is not discussed is "why downregulation of CEP31 which diminish centrosomal localization of USP9X also reduces the levels of USP9X" shown in Figure 2D right panel?
- The percentage of knock down cells of CEP131 needs to be included in Fig. 2.
- It is not clear to the reviewer why ubiquitin mutant was used to establish the type of ubiquitination. The authors should use different ubiquitin mutants including lysine 6, 11, 29, 33, 48 and 63 to establish if poly-ubiquitination of CEP131 is via lysine 48 and/or 29.
- Explanation for in vitro Deub. and Ub. assay were not clear. Did the authors use pure in vitro deub. assay using bacterial purified protein?
- Global analysis of human ubiquitin-modified proteome was used to identify site of the ubiquitination. It is not clarified how this analysis was performed, while the other problem with such in silico method is the reliability. In particular, the identified sites and mutation of these sites are not preventing totally the ubiquitination status of CEP131 (Fig. 4F). Another approach could be to perform Masspect to cnfirm or identify new sites.
- The data presented in Fig. 5 is convincing however, it is essential to include live cell imaging showing chromosomal instability in overexpressed and SiRNA treated regulated cells.
- A table is necessary to summarize the percentage of "Different" chromosomal instability phenotypes that occurs in overexpressed and down regulated cells.
- Last figure shows growth retardation cancer cells both in vitro and in vivo in ShRNA treated cells. However, in contrast to this information, previous data presented in figure Fig. 2 and Fig.5 shows no differences in the cell cycle or cell survival in siRNA treated cells. Can the authors' explain the discrepancy between these two figures?
- There are previous reports showing the involvement of other deubiquitination enzymes in chromosomal stability. Including these references might strengthen the conclusion of the present manuscript.
- Part of the discussion need to focus on or speculating how CEP131 deubiquitination is involved in genome instability of cells and what is the downstream signaling that can be affected.

Reviewer #3 (Remarks to the Author):

Review Song et al Nature Communications

In this manuscript, the authors identify CEP131 as a major interacting protein with the deubiquitinase USP9X. They provide experiments to demonstrate that CEP131 protein abundance is positively regulated by the deubiquitinase activity of USP9X. Overexpression of USP9X or CEP131 leads to centrosome amplification and genomic instability. The authors then postulate that increased levels of USP9X and CEP131 in some breast cancer cells is correlated with their metastatic potential.

Although the authors provide reasonable data for a functional link between USP9X and CEP131, the data lack some controls, the specificity of the observed phenotypes is unclear and they do not provide a compelling link between USP9X/CEP131 and breast cancer. These concerns must be adequately met for publication in Nature Communications.

Major issues:

No rescue experiments are performed to show the specificity of the RNAi triggers used. Given the propensity of off-target effects using siRNA, rescue experiments are essential. Rescue of phenotypes for critical experiments should be provided (for example, Figures 3A and D, 5C).

The specificity of the phenotypes observed upon USP9X depletion and over expression needs to be confirmed further:

1) Centriolar satellite proteins exhibit a high degree of interdependence in their localizations. For example, the loss of PCM1 or pericentrin affects the localization of CEP131 to satellites; however the loss of CEP131 does not seem to reciprocally affect these proteins (Staples CJ, JCS, 2012). It would be beneficial to show controls with additional satellite proteins to provide convincing evidence that the link between USP9X/CEP131 is direct and not a consequence of a general loss of satellite proteins. For example, is PCM1 localization disrupted in USP9X-depleted cells? does PCM1 abundance decrease after USP9X depletion?

2) The authors consistently find that expression/silencing of CEP131 only partially rescues the phenotype of depleting/over-expressing USP9X (see Figures 5 B,C and D, 7C and E). While it seems that there is a functional link between USP9X and CEP131, perhaps this indicates that USP9X functions actually functions upstream of CEP131.

Lastly, the authors try to demonstrate that elevated levels of USP9X and CEP131 are clinically relevant in a breast cancer model. I have a number of concerns about this section of the manuscript:

In panel 7A, what is the difference between the two box plots labeled 'Ma Breast 4'? Have these been mislabeled?

1) Oncomine indicates that CEP131 expression is not significantly increased in the Ma Breast 4, Finak breast and Turashvili breast samples. This argues against the authors model where increased USP9X results in increased CEP131, which itself is responsible for the observed phenotypes.

2) Relating to the correlation between the abundance of USP9X and CEP131 in cancer tissues, can the authors provide a negative control to compare. For example, does the abundance of PCM1, another satellite protein, also correlate with USP9X. Perhaps these cells have more centriolar satellites in general and therefore many of the components will correlate. I believe it is important to show that the correlation between USP9X and CEP131 is specific in these cell types.

3) Can the authors provide the parameters used for the Kaplan-Meier analyses and justification for these settings?

A) The web-based tool allows four options for survival analysis, DMFS (distant metastasis free survival), RFS (relapse free survival), OS (overall survival) and PPS (post progression survival). Using the USP9X identifier '201099_at', the survival of high and low expressors is significant for the DMFS and RFS survival, but not OS or PPS. Can the authors explain why they have only presented the DMFS and RFS survival? Is the OS and PPS not applicable for USP9X? If so, please indicate why.

B) The webtool indicates that the 'JetSet' identifier, which in the case of USP9X is 229573_at is the preferred identifier. Using this id, the difference between high and low expression is either not significant, or significant in the opposite direction (high expression correlates with better survival). Can the authors please justify why they did not use the preferred identifier?

C) Using CEP131 (id 214742_at), high expression is correlated with worse outcome for DMFS, but not RFS, OS or PPS. Since the model is that increased USP9X stabilizes CEP131, one would think that increased CEP131 expression would also result in poorer outcomes. This link only holds with the DMFS survival.

Minor Points:

The extent of CEP131 destabilization in response to USP9X depletion seems to vary between some panels. Is it possible that different siRNA durations can account for this?

Please see:

Figure 3A vs. 3D
Figure 7B

Page 7:

"elution pattern of CEP131 was largely overlapped with that of PCM1 and USP9X"

While the elution pattern of CEP131 largely overlapped with USP9X, I do not agree that it largely overlaps with PCM1. Additionally, it seems that there is a large peak of protein, as defined by the A280 trace that might elute around fraction 17. If so, it is less compelling to see multiple proteins in this fraction. This panel should be labeled more clearly to correlate the western blot to the elution profile.

Page 9:

In the domain mapping studies, the authors show that the N-terminus of USP9X is necessary for the interaction with CEP131 but do not show it is sufficient (as they do for the CEP131 domain mapping). Does the N-terminal domain alone interact with CEP131?

"USP9X truncation mutants revealed a high affinity between CEP131 and USP9X/M"

Statements of affinity cannot be made from these experiments. Please change text to reflect that this is a non-quantitative experiment.

"demonstrated CEP131 N-terminal region directly interacts with USP9X/M"

Is there a reason the authors used USP9X purified from mammalian cells? USP9X could interact another co-purifying protein that could bridge the interaction between USP9X and CEP131. Without performing experiments with purely bacterially (or at least non-mammalian) expressed protein the authors cannot claim a direct interaction.

Page 10:

The data in Figure 2B does not match the description in the text. The authors suggest that the co-localization between USP9X and Centrin is evident in S/G2, but less so in G1. However, their images are labeled G1/S and G2 so it is not possible to evaluate this claim. Can the authors provide images of G1 cells? Cyclin A staining should discriminate between G1 and S/G2.

Although PCM1 and CEP131 proteins are known to reside in centriolar satellites, very little non-centrosomal staining is apparent for these proteins. USP9X does not co-localize with the detectable satellite staining of PCM1. Can the authors comment on this observation?

Page 11:

"USP9X and CEP131 in these cells oscillated at a similar pace during"

Can the authors explain this statement? I do not observe a strong 'oscillation'. The data suggest CEP131 abundance increases around 2 hours, while USP9X increases at 4 to 6 hours. Perhaps a quantification of multiple westerns would strengthen this point. Also, at 12 hours, the phospho-H3 signal suggests cells have entered mitosis. At this point USP9X and CEP131 abundance has not decreased, yet these proteins are not detectable in mitosis in the immunofluorescence images (Figure 1A). Does the abundance of these proteins decrease by western plot at later time points?

Additionally, time of interaction between USP9X and CEP131 in Figure 2F does not seem to correlate with the initial increase in CEP131 abundance in Figure 2E.

Page 12:

Please provide biological replicates for the quantification in Figure 3D.

Does the loss of CEP131 from the centrosome specific in USP9X siRNA conditions? i.e. are other components also displaced (other than centrin)? Possibly other satellite components such as CEP290 or PCM1? The loss of CEP131 from centrosomes could represent a general loss of satellite components.

Figure 3F. Please provide quantification for this. The CEP131 abundance in MDA-MB-231, ECC1 and Ishikawa cells look reasonably similar while having noticeably different levels of USP9X. Perhaps a correlation plot would be more convincing.

Page 14:

What is FLAG-tagged in Figure 4D? Is it FLAG-USP9X? Please indicate.

The level of overexpression of the wt and C1566S forms of USP9X is not comparable. Is it possible to use a higher doxycycline concentration for the C1566S protein?

Page 15:

The labels on Figure 4F are confusing. The bottom panel is labeled CEP131/K504, yet CEP131/K254R is indicated as being present in one of the lanes.

Please include biological replicates for the quantification of Figure 4G.

Page 16:

Do the FACS profiles correspond to the cells analyzed in the immunofluorescence panels in Figure 5A and C? If so, the profiles look like mostly G1 cells. Can the authors provide A) a FACS profile of cycling cells to use for a comparison B) another method, such as cyclin A staining or western blot

to demonstrate the cells are in the expected cell cycle stage.

For Figures 5B and C it would be nice to see western blots to judge protein abundance. For example, USP9X over expression should stabilize CEP131, but CEP131 siRNA should deplete the protein - it is necessary to judge what the final steady state level of the protein is in this case to interpret the functional data.

Page 18:

Can Figure 6C be rearranged so all normal and all tumor lines are together?

Page 19:

In Figure 7C, shRNA against CEP131 results in a weaker colony formation phenotype than using shRNA against USP9X. However, the abundance of CEP131 is lower using shRNA against CEP131 than in cells using USP9X shRNA. This is not entirely consistent with the model where USP9X regulates CEP131 stability.

Perhaps some of the data that is replicated in different cell lines can be presented in Supplementary information.

page7:

excessive FLAG peptides: excess FLAG peptide(s)
centriolar satellite protein CEP131 was also identified: were also identified

Page 10:

deubiquitinase US9X: USP9X

Page 19:

samples (Fig. 6D, upper panle) and: upper panel

Summary of changes

1. The mechanism of USP9X promoted-centrosome amplification has been investigated and the results point to a role of CEP131-regulated CDK2 localization in USP9X-promoted centrosome biogenesis (Figure S6E, Figure S6F and Figure S6G).
2. Co-immunoprecipitation assays were performed to analyze association of USP9X with the other interactors identified in our USP9X-pull down experiments. The results validated ITCH, IPO5, PRMT5, and PPM1B as USP9X interactors (Figure S1A), but the protein abundance of IPO5, PRMT5, and PPM1B are not subjected to USP9X regulation (Figure S10A).
3. The experiments in Figure 3B, Figure 4A and Figure 4B have been optimized and repeated and the data have been replaced, and the experiments in previous Figure 3D have been repeated and results from biological triplicate experiments have been shown in Figure 3D, Figure S3C, and Figure S3D.
4. We have tested the effects of other USP9X substrates on USP9X regulated centrosome amplification and breast carcinogenesis. The results provided in Figure S10 suggest that USP9X-promoted CEP131 stabilization functions in centrosome biogenesis and breast cancer cell survival in an ITCH and MCL1 independent manner.
5. The function and role of USP9X in HMECs (human mammary gland epithelial cells) have been investigated and the results provided in Figure S3B and Figure S6C indicated that USP9X plays a conserved role in non-cancerous cells and cancer cells.
6. We utilized CRISPR/Cas9 system to knockout USP9X in HMECs and demonstrated that in USP9X deficient cells the expression level of CEP131 is downregulated and the downregulation of CEP131 in these cells could be rescued by forced expression of wild type USP9X, but not USP9X mutant (USP9X/C1566S) (Figure S5A), while the mRNA expression level of CEP131 was not altered (Figure S5A).
7. Original Figure 1E has been moved to SUPPLEMENTAL FIGURES and shown as Figure S1B; According to the suggestion of Reviewer #3, Figure 3A, Figure 3D, Figure 4C and Figure 5B have been moved to SUPPLEMENTAL FIGURES and shown as Figure S3A, Figure S3D, Figure S5B and Figure S6B, respectively.
8. The y axis in the right panel of Figure 2D and Figure 3F has been replaced with new label “Relative USP9X intensity in centrosome” and “Relative CEP131 intensity in centrosome”.

9. The percentage of CEP131 or USP9X knockdown cells has been provided in Figure 2D and Figure 3F, respectively.
10. *In vivo* deubiquitination assays have been performed with ubiquitin mutant with all lysine residues replaced by arginine except K29 (K29-only) or K48 (K48-only) or K63 (K63-only) to differentiate ubiquitin species opposed by USP9X on poly-ubiquitinated CEP131. The results provided in Figure S5D indicated K48-linked ubiquitin species are the major forms targeted by USP9X.
11. Mass spectrometry analysis of ubiquitin conjugation sites on CEP131 is provided in Figure S5E.
12. Live cell imaging analysis of chromosome instability has been provided in Figure S7 and Supplementary Movie 1.
13. The effect of USP9X-promoted CEP131 stabilization on genome stability and cell death has been investigated and the results have been provided in Figure S9.
14. Previous reports on the involvement of other deubiquitination enzymes in chromosomal stability have been included/cited in the revision (paragraph 1, page 30).
15. How CEP131 deubiquitination is involved in centrosome amplification and genome instability and what is the downstream signaling that can be affected have been discussed (paragraph 2, page 28).
16. Rescue experiments have been performed in USP9X deficient cells and the results have been provided in Figure 3E, Figure S6D and Figure S10C.
17. The effects of USP9X depletion on the localization and abundance of PCM1, CEP290 and Pericentrin have been examined and the results have been shown in Figure S4.
18. The second dataset of 'Ma Breast 4' (Figure 6A) has been removed.
19. The expression level of PCM1 and CEP290 in breast cancer samples and tumor adjacent tissues has been examined and the results have been provided in Figure S8A and Figure S8B.
20. Figure 6E of *Kaplan-Meier* analysis has been removed in the revision.
21. Kaplan-Meier survival analyses in original Figure 6E have been removed.

22. Description of FPLC results (Figure 1D) in the text has been re-worded.
23. Description of pull-down between USP9X truncation mutants and CEP131 (Figure 1E) has been re-worded.
24. USP9X deletion mutants purified from Sf9 cells have been used to re-perform the pull-down experiments and the data has been provided in Figure 1F and Figure 1G.
25. Immunostaining with antibodies against USP9X and Centrin with cyclin A-CFP stably expressing U2OS cells has been provided in Figure S2A.
26. Biological triplicate experiments corresponding to original Figure 3D have been provided in Figure 3D, Figure S3C and Figure S3D, and Biological triplicate experiments corresponding to original Figure 4G have been provided in Figure 4G and Figure S5F. The quantitation and statistical analysis of these results have been provided in Figure 3D, Figure S3D and Figure 4G.
27. The results with Western blotting analysis judging the abundance of CEP131 and USP9X post mitosis, have been shown in Figure S2D.
28. Biological triplicate experiments corresponding to original Figure 2E together with quantification and correlation of USP9X and CEP131 has been provided in Figure 2E and Figure S2C.
29. Myc-tagged and FLAG-tagged proteins have been relabeled in Figure 4C and Figure 4D.
30. Higher concentration of doxycycline was used to induce C1566S expression and the deubiquitination assay has been re-performed as shown in Figure S5C.
31. Figure 4F has been relabeled as indicated.
32. FACS profiles and Western blotting results corresponding to synchronized cells used in Figure 5A have been provided in Figure S6A.
33. FACS profiles corresponding to control or HU treated cells have been provided in Figure 5C.
34. The results with Western blotting analysis judging protein abundance in immunofluorescence stainings have been provided in Figure S6A, Figure 5B, Figure

5C, Figure 5D, Figure S4A, Figure S4C, Figure S6B, Figure S6C, Figure S6D, Figure S6E, Figure S6F, Figure S6G, Figure S10D and Figure S10E as indicated.

35. The misspelled or excess words pointed by Reviewer #3 have been corrected in the revision.

36. Two additional references relevant to USP9X physiological or pathological function are cited in the manuscript (paragraph 1, page 6).

Oishi S, et al. Usp9x-deficiency disrupts the morphological development of the postnatal hippocampal dentate gyrus. *Sci Rep* 6, 25783 (2016).

Paemka L, et al. Seizures are regulated by ubiquitin-specific peptidase 9 X-linked (USP9X), a de-ubiquitinase. *PLoS Genet* 11, e1005022 (2015).

37. The description of all new results has been incorporated into the appropriate position in the revised manuscript as indicated. All of the changes in the revision text are annotated in the Word file.

38. SUPPLEMENTAL FIGURES and SUPPLEMENTAL FIGURE LEGENDS are provided in the revision.

Response to Reviewer #1's comments-

Reviewer #1: *The first difficulty of the paper is that it attempts to give a role to USP9X in regulating CEP131 function when we do not understand what CEP131 - or indeed centriolar satellites - actually do. The authors set out to study USP9X partners and identify CEP131. Unfortunately they do not really indicate in full the other proteins that they identify as interactors and there must be many. Part of the difficulty appears to be that they present data for gel purified interacting proteins rather than presenting unbiased analysis of all interactors identified in the pull down. This is important particularly in interpreting the subsequent experiments in which USP9X function is removed. I am certain that they are correct and that CEP131 is one of USP9X's partners - they present good evidence for this in Fig. 1 - but what are ALL its other partners? Nevertheless, an association with centrosomes (Fig. 2) makes sense if USP9X is a CEP131 partner.*

Authors: The cellular function of CEP131 and centriolar satellites remains to be delineated, as the reviewer rightfully points out. A recent study reported that CEP131 is involved in centrosome duplication via regulating centrosomal localization of CDK2¹, a cyclin-dependent kinase with an established role in centrosome biogenesis^{2, 3}. This is consistent with our finding that USP9X-promoted CEP131 stabilization is required for centrosome biogenesis. In responding to the reviewer's criticism, we examined the impact of USP9X loss-of-function on CDK2 localization in centrosome and found that, similar to CEP131 depletion, USP9X deficiency was associated with an impaired centrosomal localization of CDK2 (Figure S6E), while the protein level of CDK2 remained unchanged in both USP9X and CEP131 knockdown cells (Figure S6E). Importantly, USP9X depletion-induced phenotype of CDK2 localization could be rescued by CEP131 overexpression (Figure S6F). Moreover, USP9X promoted-centrosome amplification was abrogated upon CDK2 depletion (Figure S6G). These results point to a role of CEP131-regulated CDK2 localization in USP9X-promoted centrosome biogenesis.

Our affinity purification and mass spectrometry analysis showed that USP9X is associated with multiple proteins (Figure 1A). As stated in the manuscript, USP9X has been reported to target several cytosolic proteins, and its association and potential targeting of CEP131 was particularly interesting as the cellular compartmentalization and biological function of CEP131 are distinct from that of the reported substrates of USP9X. Thus, we focused our study on the regulation of CEP131. In responding to the reviewer's

criticism, co-immunoprecipitation assays were performed to analyze the other interactors of USP9X identified in our pull down experiments. The results validated ITCH, IPO5, PRMT5, and PPM1B as USP9X interactors (Figure S1A).

Reviewer #1: The effects shown in subsequent figures and used to reach conclusions about USP9X function are in many cases modest. For example, the loss of CEP131 following USP9X RNAi in MCF7 cells is not convincing (Fig. 3B). Moreover, the change in CEP131 half-life following USP9X RNAi particularly in U2OS cells is certainly not dramatic. Similarly, the increase in CEP131 following expression of elevated USP9X is not dramatic (Fig. 4A), as claimed, and neither are the diminution of CEP131 levels following inhibition of the DUB with WP1130 (Fig. 4D).

Authors: In responding to the reviewer's criticism, the experiments in Figure 3B, Figure 4A and Figure 4B (inhibition of the DUB with WP1130) have been optimized, including improving the knockdown or overexpression efficiency and extending the time of doxycycline or inhibitor treatment to cells, and repeated, and the data have been replaced. The experiments in previous Figure 3D (CHX-chase assay) have been repeated and results from biological triplicate experiments have been shown in Figure 3D, Figure S3C, and Figure S3D.

Reviewer #1: It seems quite likely that CEP131 is a USP9Z substrate but the question is what are its other substrates and how might these affect, not only the centrosome cycle, but more broadly oncogenesis. Without this knowledge it is difficult to interpret the meanings of Figures 6 and 7 that largely analyse correlative findings. In conclusion, much more needs to be done to understand the roles of CEP131 in centrosome biology and the full range of substrates of UPS9X before it will be possibly to correctly interpret this study.

Authors: In responding to the reviewer's comments, we have performed a series of experiments with other substrates of USP9X: 1) In agreement with previous reports^{4, 5, 6, 7, 8}, we found that USP9X knockdown did result in decreased expression of ITCH (Figure S10A), an E3 ligase involved in carcinogenesis^{4, 6}, and of MCL1, an anti-apoptotic regulator implicated in cancer^{9, 10, 11, 12}. However, USP9X, but not CEP131, could be co-immunoprecipitated by ITCH or MCL1 (Figure S10B); 2) Colony formation assays demonstrated that while overexpression of ITCH or MCL1 could rescue the growth inhibitory phenotype resulted from USP9X depletion to certain extent as CEP131 did, simultaneous expression of CEP131 and ITCH or MCL1 showed an additive effect (Figure S10C); 3) while loss of USP9X-associated defects of centrosome amplification could be rescued by CEP131 overexpression (Figure S10D), overexpression of ITCH or MCL1 could not rescue the phenotype (Figure S10D); 4) knockdown of either ITCH or MCL1 had no effect on centrosomal localization of USP9X and CEP131 (Figure S10E);

and 5) Western blotting analysis of cellular lysates of USP9X-deficient MCF-7 cells revealed that the expression of IPO5, PRMT5 and PPM1B, which are also identified as interactors of USP9X in our experiments (Figure S1A), was essentially unchanged (Figure S10A), suggesting that the protein abundance of IPO5, PRMT5 and PPM1B is not subjected to USP9X regulation, although the effect of these proteins on USP9X functionality needs to be further investigated. The data have been added to the revision as indicated.

References for Reviewer #1

1. Kodani A, *et al.* Centriolar satellites assemble centrosomal microcephaly proteins to recruit CDK2 and promote centriole duplication. *eLife* **4**, (2015).
2. Matsumoto Y, Hayashi K, Nishida E. Cyclin-dependent kinase 2 (Cdk2) is required for centrosome duplication in mammalian cells. *Current biology : CB* **9**, 429-432 (1999).
3. Adon AM, *et al.* Cdk2 and Cdk4 regulate the centrosome cycle and are critical mediators of centrosome amplification in p53-null cells. *Molecular and cellular biology* **30**, 694-710 (2010).
4. Salah Z, Itzhaki E, Aqeilan RI. The ubiquitin E3 ligase ITCH enhances breast tumor progression by inhibiting the Hippo tumor suppressor pathway. *Oncotarget* **5**, 10886-10900 (2014).
5. Mouchantaf R, Azakir BA, McPherson PS, Millard SM, Wood SA, Angers A. The ubiquitin ligase itch is auto-ubiquitylated in vivo and in vitro but is protected from degradation by interacting with the deubiquitylating enzyme FAM/USP9X. *J Biol Chem* **281**, 38738-38747 (2006).
6. Perez-Mancera PA, *et al.* The deubiquitinase USP9X suppresses pancreatic ductal adenocarcinoma. *Nature* **486**, 266-270 (2012).
7. Schwickart M, *et al.* Deubiquitinase USP9X stabilizes MCL1 and promotes tumour cell survival. *Nature* **463**, 103-107 (2010).
8. Trivigno D, Essmann F, Huber SM, Rudner J. Deubiquitinase USP9x confers radioresistance through stabilization of Mcl-1. *Neoplasia* **14**, 893-904 (2012).
9. Gao J, *et al.* MiR-26a inhibits proliferation and migration of breast cancer through repression of MCL-1. *PloS one* **8**, e65138 (2013).

10. Inuzuka H, *et al.* SCF(FBW7) regulates cellular apoptosis by targeting MCL1 for ubiquitylation and destruction. *Nature* **471**, 104-109 (2011).
11. Wertz IE, *et al.* Sensitivity to antitubulin chemotherapeutics is regulated by MCL1 and FBW7. *Nature* **471**, 110-114 (2011).
12. Yang L, *et al.* Wnt modulates MCL1 to control cell survival in triple negative breast cancer. *BMC cancer* **14**, 124 (2014).

Response to Reviewer #2's comments-

Major questions:

Reviewer #2 - *What is the function and role of USP9X in non-cancerous cells? If the answer is that the expression of USP9X is low in normal cells, this is not clear by the data shown in Fig. 3F.*

Authors: USP9X was co-localized with Centrin at centrosome in non-cancerous cells HMECs (human mammary gland epithelial cells) (Figure S2B), and the protein, but not mRNA expression level of CEP131 was down-regulated upon USP9X depletion in HMECs (Figure S3B). Consistent with the role of USP9X in cancerous cells, we demonstrated that USP9X promotes centrosome biogenesis in a CEP131-dependent manner in HMECs (Figure S6C).

Our data indicate that the expression of USP9X is low in non-cancerous cells compared to cancerous cells of the same tissue origin (original Figure 3F, now shown as Figure 3G). Consistently, measurement of the expression of USP9X and CEP131 in human tissues by Western blotting or immunohistological analysis showed the protein levels of both USP9X and CEP131 were substantially elevated in breast carcinoma samples (Figure 6C and Figure 6D).

Reviewer #2 - *USP9X transgenic animal has been generated previously. Including experiments using USP9X primary cells isolated from knockout animals can strength the conclusion of the authors. This important since all of the experiments present in the manuscript are performed by overexpressing/siRNA treating human cancer cell lines.*

Authors: We appreciate the reviewer for this comment. However, we have tried to obtain USP9X transgenic animals from the researchers who generated the animals^{1, 2} without success. To address the issue, we utilized CRISPR/Cas9 system to knockout USP9X in HMECs and demonstrated that, in USP9X deficient normal mammary cells, the expression level of CEP131 decreased and the downregulation of CEP131 in these cells could be rescued by forced expression of wild type USP9X, but not USP9X mutant (USP9X/C1566S) (Figure S5A), while the mRNA expression level of CEP131 was not altered (Figure S5A). These results are consistent with our observations in the manuscript.

Reviewer #2 - *Fig. 1E is not necessary and can be presented in Suppl. figures since the same information is presented in 1D.*

Authors: Figure 1E has been moved to SUPPLEMENTAL FIGURES and shown as Figure S1B.

Reviewer #2 - *The authors need to confirm direct interaction between USP9X and CEP131 using purified proteins. Fig. 1G shows the binding between "truncated mutant" of USP9X and CEP131.*

Authors: We agree with the reviewer on this point. However, possibly due to its high molecular mass (~300 kDa), USP9X full length protein is difficult to purify in either bacteria or insect Sf9 cells, despite our intensive efforts. To address the reviewer's concern, USP9X deletion mutants purified from insect Sf9 cells have been used to repeat the pull-down experiments and the data has been provided in Figure 1F and Figure 1G.

Reviewer #2 - *Downregulation of USP9X leads to reduced levels of CEP131. What is not discussed is "why downregulation of CEP31 which diminish centrosomal localization of USP9X also reduces the levels of USP9X" shown in Figure 2D right panel?*

Authors: The y axis labeled with "Relative USP9X intensity" represents the relative USP9X intensity in centrosome, not in the whole cell. We apologize for the confusion and have changed the labeling of y axis to "Relative USP9X intensity in centrosome" (Figure 2D right panel). As demonstrated in the right lower panel of Figure 7B and the right upper panel of Figure 7D, CEP131 downregulation had no effect on the expression of USP9X.

Reviewer #2 - *The percentage of knock down cells of CEP131 needs to be included in Fig. 2.*

Authors: The percentage of knockdown cells of CEP131 has been included in Figure 2D.

Reviewer #2 - *It is not clear to the reviewer why ubiquitin mutant was used to establish the type of ubiquitination. The authors should use different ubiquitin mutants including lysine 6, 11, 29, 33, 48 and 63 to establish if poly-ubiquitination of CEP131 is via lysine 48 and/or 29.*

Authors: The ubiquitin mutant with all lysine residues replaced by arginine (Ub/mt) was used as a negative control in poly-ubiquitin chain detection. To comply with the reviewer, we have added additional ubiquitin mutants with all lysine residues replaced by arginine except K29 (K29-only) or K48 (K48-only) or K63 (K63-only) to differentiate ubiquitin

species opposed by USP9X on poly-ubiquitinated CEP131. The results in Figure S5D indicate that K48-linked ubiquitin species are the major forms of CEP131 targeted by USP9X.

Reviewer #2 - *Explanation for in vitro Deub. and Ub. assay were not clear. Did the authors use pure in vitro deub. assay using bacterial purified protein?*

Authors: As explained above, full length USP9X protein is difficult to purify. Thus, we used FLAG-USP9X/wt and FLAG-USP9X/C223S purified from mammalian cells in high salt and detergent buffer for *in vitro* Deub assays. Moreover, since the *bona fide* E3 ligase for CEP131 is currently unidentified and Ub-conjugated CEP131 could not be generated *in vitro*, we retrieved HA-Ub-conjugated FLAG-CEP131 from HeLa cells with tandem purification using anti-FLAG and anti-HA affinity gel in high salt and detergent buffer. The information has been clarified in *In Vitro* Deubiquitination Assay (Supplemental Methods).

Reviewer #2 - *Global analysis of human ubiquitin-modified proteome was used to identify site of the ubiquitination. It is not clarified how this analysis was performed, while the other problem with such in silico method is the reliability. In particular, the identified sites and mutation of these sites are not preventing totally the ubiquitination status of CEP131 (Fig. 4F). Another approach could be to perform Masspect to confirm or identify new sites.*

Authors: Ubiquitination sites were identified by SILAC-based quantitative analysis of human ubiquitin-modified proteome³. Detailed information on 19,000 diGly-modified lysine residues within 5000 proteins including CEP131 is shown in Table S2 of the reference³.

Although K254R did not totally abolish CEP131 ubiquitination, this mutation rendered CEP131 resistant to USP9X deubiquitination (Figure 4F) and the half-life of CEP131/K254R was not affected upon USP9X depletion (Figure 4G and Figure S5F), indicating that K254 residue of CEP131 is the major poly-ubiquitin targeting site opposed by USP9X. It is possible that other ubiquitin conjugating sites exist on CEP131, which are catalyzed or opposed by other E3 ligases or deubiquitinases, respectively.

In responding to the reviewer's suggestion, we have employed mass spectrometry to analyze ubiquitin conjugation sites on CEP131 and found two CEP131 peptides carrying ubiquitin-modified sites at lysine residues 254 and 714 (Figure S5E).

Reviewer #2 - *The data presented in Fig. 5 is convincing however, it is essential to include live cell imaging showing chromosomal instability in overexpressed and SiRNA*

treated regulated cells.

Authors: To comply with the reviewer's requests, we have provided live cell imaging in Figure S7 and Supplementary Movie 1.

Reviewer #2 - *A table is necessary to summarize the percentage of "Different" chromosomal instability phenotypes that occurs in overexpressed and down regulated cells.*

Authors: Such a table has been provided in Figure 5D.

Reviewer #2 - *Last figure shows growth retardation cancer cells both in vitro and in vivo in ShRNA treated cells. However, in contrast to this information, previous data presented in figure Fig. 2 and Fig.5 shows no differences in the cell cycle or cell survival in siRNA treated cells. Can the authors' explain the discrepancy between these two figures?*

Authors: Cell cycle profiles but not cell survivals were examined in Figure 2E, original Figure 5A and original Figure 5C. Our experiments indicate that USP9X/CEP131-promoted breast cancer cell growth (Figure 7) is not through cell cycle control, as either overexpression or knockdown of USP9X/CEP131 had minimal effect on cell cycle progression (Figure S6A and Figure 5C). Since centrosome dysregulation-associated mitotic defects could result in genome instability and cell apoptosis^{4,5,6}, we examined whether USP9X-promoted CEP131 stabilization plays a role in genome instability and cell apoptosis. Indeed, we demonstrated either USP9X or CEP131 depletion resulted in markedly accumulation of γ H2AX (Figure S9A) and severe apoptosis of MCF-7 cells (Figure S9B). Moreover, USP9X depletion-associated phenotypes could be alleviated by CEP131 overexpression (Figure S9B). These results indicate that USP9X/CEP131-promoted breast cancer cell survival is through controlling cell apoptosis.

Reviewer #2 - *There are previous reports showing the involvement of other deubiquitination enzymes in chromosomal stability. Including these references might strengthen the conclusion of the present manuscript.*

Authors: Previous reports on the involvement of other deubiquitination enzymes in chromosomal stability have been included/cited in the revision.

Reviewer #2 - *Part of the discussion need to focus on or speculating how CEP131 deubiquitination is involved in genome instability of cells and what is the downstream signaling that can be affected.*

Authors: We have added the relevant discussion in the revision.

References for Reviewer #2

1. Stegeman S, *et al.* Loss of Usp9x disrupts cortical architecture, hippocampal development and TGFbeta-mediated axonogenesis. *PloS one* **8**, e68287 (2013).
2. Oishi S, *et al.* Usp9x-deficiency disrupts the morphological development of the postnatal hippocampal dentate gyrus. *Scientific reports* **6**, 25783 (2016).
3. Kim W, *et al.* Systematic and quantitative assessment of the ubiquitin-modified proteome. *Molecular cell* **44**, 325-340 (2011).
4. Staples CJ, *et al.* The centriolar satellite protein Cep131 is important for genome stability. *Journal of cell science* **125**, 4770-4779 (2012).
5. Yoshino Y, Ishioka C. Inhibition of glycogen synthase kinase-3 beta induces apoptosis and mitotic catastrophe by disrupting centrosome regulation in cancer cells. *Scientific reports* **5**, 13249 (2015).
6. Kimura M, Yoshioka T, Saio M, Banno Y, Nagaoka H, Okano Y. Mitotic catastrophe and cell death induced by depletion of centrosomal proteins. *Cell death & disease* **4**, e603 (2013).

Response to Reviewer #3's comments-

Major issues:

Reviewer #3: *No rescue experiments are performed to show the specificity of the RNAi triggers used. Given the propensity of off-target effects using siRNA, rescue experiments are essential. Rescue of phenotypes for critical experiments should be provided (for example, Figures 3A and D, 5C).*

Authors: In responding to the reviewer's concerns, we have performed the following experiments: 1) Control U2OS cells or U2OS cells stably expressing USP9X was transfected with siRNA targeting 3'UTR of *USP9X* mRNA. Western blotting analysis revealed that USP9X overexpression was able to restore the expression of CEP131 (Figure 3E) and prolong the half-life of CEP131 (Figure 3E) in USP9X deficient cells; and 2) Control U2OS cells or U2OS cells stably expressing USP9X was transfected with siRNA targeting 3'UTR of *USP9X* mRNA, and immunostaining or colony formation analysis revealed that the centrosome amplification defect or growth inhibitory effect induced by USP9X depletion was overcome by USP9X overexpression, respectively (Figure S6D and Figure S10C).

Reviewer #3: *The specificity of the phenotypes observed upon USP9X depletion and over expression needs to be confirmed further:*

1) Centriolar satellite proteins exhibit a high degree of interdependence in their localizations. For example, the loss of PCMI or pericentrin affects the localization of CEP131 to satellites; however the loss of CEP131 does not seem to reciprocally affect these proteins (Staples CJ, JCS, 2012). It would be beneficial to show controls with additional satellite proteins to provide convincing evidence that the link between USP9X/CEP131 is direct and not a consequence of a general loss of satellite proteins. For example, is PCMI localization disrupted in USP9X-depleted cells? does PCMI abundance decrease after USP9X depletion?

Authors: To comply with the reviewer's requests, we have examined the localization and abundance of PCMI after USP9X depletion. The results demonstrated that the centrosomal localization of PCMI was mildly interrupted (Figure S4A) and the protein, but not mRNA, level of PCMI decreased (Figure S4B) upon USP9X depletion, indicating that PCMI is a potential substrate of USP9X. However, we noted that, in USP9X-deficient cells, CEP131 could be effectively recruited to centrosome (Figure

S4C), suggesting that mild disruption of PCM1 centrosomal localization associated with USP9X depletion has limited effect on CEP131 recruitment. Nevertheless, we agree that how PCM1 contributes to USP9X-regulated centrosome biogenesis remains to be investigated in the future.

To further exclude the possibility that CEP131 or PCM1 dis-localization from centrosome is indirectly affected by other centrosome components, we examined the localization and expression level of Pericentrin and CEP290, both of which were critical satellite components and reported to be essential for the centrosomal restriction of PCM1 and CEP131¹. The results in Figure S4A and Figure S4B indicate that the localization and abundance of these proteins were unaffected upon USP9X knockdown.

These observations, together with our findings that USP9X physiologically interacts with CEP131 and promotes CEP131 deubiquitination *in vitro* and *in vivo*, support a notion that the link between USP9X and CEP131 is direct and USP9X depletion-associated loss of CEP131 from centrosome is not a consequence of loss of general satellite proteins.

Reviewer #3: 2) *The authors consistently find that expression/silencing of CEP131 only partially rescues the phenotype of depleting/over-expressing USP9X (see Figures 5 B, C and D, 7C and E). While it seems that there is a functional link between USP9X and CEP131, perhaps this indicates that USP9X functions actually functions upstream of CEP131.*

Authors: We agree with the reviewer' point that USP9X functions upstream of CEP131. It has been reported that USP9X targets, in addition to CEP131, other cellular proteins including ITCH and MCL1^{2,3,4}. This might explain why expression/silencing of CEP131 only partially rescues the phenotype of depleting/over-expressing USP9X. Moreover, technical limitations such as the efficiency of overexpression/silencing may also contribute to the effect of partial rescue. These points have been discussed in the revision.

Reviewer #3: *Lastly, the authors try to demonstrate that elevated levels of USP9X and CEP131 are clinically relevant in a breast cancer model. I have a number of concerns about this section of the manuscript:*

In panel 7A, what is the difference between the two box plots labeled 'Ma Breast 4'? Have these been mislabeled?

Authors: In the second dataset of 'Ma Breast 4', the second group labeled as number 2 (original Figure 6A) represents ductal breast carcinoma in situ of stroma, not epithelia cells. We apologize for the inappropriate representation and have removed the data.

Reviewer #3: 1) *Oncomine indicates that CEP131 expression is not significantly*

increased in the Ma Breast 4, Finak breast and Turashvili breast samples. This argues against the author's model where increased USP9X results in increased CEP131, which itself is responsible for the observed phenotypes.

Authors: Oncomine database concerns profiling gene expression in the level of mRNA, not protein. Our overall argument is that the protein, but not mRNA, abundance of CEP131 is regulated by USP9X and elevated in breast carcinomas. We have clarified the issue in the revision.

Reviewer #3: *2) Relating to the correlation between the abundance of USP9X and CEP131 in cancer tissues, can the authors provide a negative control to compare. For example, does the abundance of PCM1, another satellite protein, also correlate with USP9X. Perhaps these cells have more centriolar satellites in general and therefore many of the components will correlate. I believe it is important to show that the correlation between USP9X and CEP131 is specific in these cell types.*

Authors: To comply with the reviewer's requests, we have examined the expression level of PCM1 in breast cancer samples and tumor adjacent tissues. As demonstrated in Figure S8A and Figure S8B, the protein abundance of PCM1 was elevated in breast cancer and correlated with that of USP9X. This observation is consistent with the finding that PCM1 is a candidate substrate of USP9X. However, the expression of CEP290, another essential satellite protein, was not elevated in breast cancer, nor correlated with that of USP9X (Figure S8A and Figure S8B).

Reviewer #3: *3) Can the authors provide the parameters used for the Kaplan-Meier analyses and justification for these settings?*

A) The web-based tool allows four options for survival analysis, DMFS (distant metastasis free survival), RFS (relapse free survival), OS (overall survival) and PPS (post progression survival). Using the USP9X identifier '201099_at', the survival of high and low expressors is significant for the DMFS and RFS survival, but not OS or PPS. Can the authors explain why they have only presented the DMFS and RFS survival? Is the OS and PPS not applicable for USP9X? If so, please indicate why.

B) The webtool indicates that the 'JetSet' identifier, which in the case of USP9X is 229573_at is the preferred identifier. Using this id, the difference between high and low expression is either not significant, or significant in the opposite direction (high expression correlates with better survival). Can the authors please justify why they did not use the preferred identifier?

C) Using CEP131 (id 214742_at), high expression is correlated with worse outcome for DMFS, but not RFS, OS or PPS. Since the model is that increased USP9X stabilizes CEP131, one would think that increased CEP131 expression would also result in poorer outcomes. This link only holds with the DMFS survival.

Authors:

A) Unlike DMFS and RFS, the OS survival difference of high and low expression of *USP9X* is not significant in all of the *USP9X* identifiers. Since the samples enlisted in Kaplan-Meier plotter for OS survival include multiple types or stages of breast cancer with different treatments, we have tried to stratify the samples into different categories. However, the analysis in all sub-categories is not convincing due to sample size limitation. Thus, we did not provide these results. Similarly, patient number enlisted into PPS analysis is less than 200. Thus, we did not present these data.

B) Although the 'JetSet' identifier of *USP9X* is the preferred Affymetrix probe, the cohort with *USP9X* identifier '201099_at' that we presented in the manuscript contains more than two times of patients (3554) compared to that in the cohort with *USP9X* 'JetSet' identifier (1660). Thus, we did not use the preferred identifier.

C) In this study, we demonstrated that *USP9X* regulates the expression of *CEP131* at post-translational level through the deubiquitinase activity of *USP9X*, while the survival information retrieved from K-M plotter is based on mRNA level. Whether and how the abundance of *CEP131* protein is correlated with the outcome for RFS, OS or PPS remain to be investigated.

We appreciate the reviewer for these points. To avoid confusions, these data have been removed in the revision.

Minor Points:

Reviewer #3: *The extent of CEP131 destabilization in response to USP9X depletion seems to vary between some panels. Is it possible that different siRNA durations can account for this? Please see: Figure 3A vs. 3D Figure 7B*

Authors: We admit this and agree with the reviewer's point. The transfection duration for Figure 3A and Figure S3A were 96 hours, while that in Figure 3D, Figure S3C and Figure S3D were about 110 hours. Chemically synthesized siRNAs were used in Figure 3A, Figure 3D, Figure S3A and Figure S3D in MCF-7 or U2OS cells, while lentiviruses carrying shRNA stably integrated into MCF-7 cells were used in Figure 7B. We hope the reviewer agree that the overall observation that *USP9X* depletion was associated with *CEP131* destabilization can still hold.

Reviewer #3: *Page 7: "elution pattern of CEP131 was largely overlapped with that of PCMI and USP9X" While the elution pattern of CEP131 largely overlapped with USP9X,*

I do not agree that it largely overlaps with PCM1. Additionally, it seems that there is a large peak of protein, as defined by the A280 trace that might elute around fraction 17. If so, it is less compelling to see multiple proteins in this fraction. This panel should be labeled more clearly to correlate the western blot to the elution profile.

Authors: We agree with the reviewer's point that the elution pattern of CEP131 largely overlapped with that of USP9X, but not with that of PCM1, and thus we have modified the statement (the last paragraph, Page 8) in the revision. In addition, we have carefully checked the fraction labels of FPLC elution profiles and Western blotting. The counting for fraction number in silver staining or Western blotting began when samples were collected, which is different from that in A280 trace. The largest peak of protein fraction is the excess 3 × FLAG peptides (original Figure 1E, now shown as Figure S1B).

Reviewer #3: *Page 9: In the domain mapping studies, the authors show that the N-terminus of USP9X is necessary for the interaction with CEP131 but do not show it is sufficient (as they do for the CEP131 domain mapping). Does the N-terminal domain alone interact with CEP131?*

Authors: Our data indicate that the middle region, not the N-terminus, of USP9X is necessary for the interaction with CEP131. As demonstrated in the lower panel of Figure 1E and in Figure 1F, the middle region of USP9X (USP9X/M) is sufficient for USP9X interaction with CEP131.

Reviewer #3: *"USP9X truncation mutants revealed a high affinity between CEP131 and USP9X/M" Statements of affinity cannot be made from these experiments. Please change text to reflect that this is a non-quantitative experiment.*

Authors: The text has been modified in the revision.

Reviewer #3: *"demonstrated CEP131 N-terminal region directly interacts with USP9X/M" Is there a reason the authors used USP9X purified from mammalian cells? USP9X could interact another co-purifying protein that could bridge the interaction between USP9X and CEP131. Without performing experiments with purely bacterially (or at least non-mammalian) expressed protein the authors cannot claim a direct interaction.*

Authors: To address the reviewer's concern, USP9X deletion mutants purified from insect Sf9 cells have been used to repeat the pull-down experiments and the data has been provided in Figure 1F and Figure 1G.

Reviewer #3: *Page 10: The data in Figure 2B does not match the description in the text.*

The authors suggest that the co-localization between USP9X and Centrin is evident in S/G2, but less so in G1. However, their images are labeled G1/S and G2 so it is not possible to evaluate this claim. Can the authors provide images of G1 cells? Cyclin A staining should discriminate between G1 and S/G2.

Authors: The images of cells in G₁ and S phases have been provided in Figure S2A.

Reviewer #3: Although PCM1 and CEP131 proteins are known to reside in centriolar satellites, very little non-centrosomal staining is apparent for these proteins. USP9X does not co-localize with the detectable satellite staining of PCM1. Can the authors comment on this observation?

Authors: To clearly substantiate the centrosomal signal of PCM1, CEP131 and USP9X, we performed immunofluorescence assay with antibodies against PCM1, CEP131 and USP9X in high dilution factors 1:500, 1:1000 and 1:500, respectively.

After contrast adjustment of the images in Figure 2C, the overlapping signal for PCM1 and USP9X is almost as strong as that of CEP131 and USP9X. To further address the reviewer's concern, we performed immunostainings with antibodies against USP9X and PCM1, CEP290 or Pericentrin and demonstrated that USP9X is co-localized with satellite staining of these satellite components (Figure S4A).

Reviewer #3: Page 11: "USP9X and CEP131 in these cells oscillated at a similar pace during" Can the authors explain this statement? I do not observe a strong 'oscillation'. The data suggest CEP131 abundance increases around 2 hours, while USP9X increases at 4 to 6 hours. Perhaps a quantification of multiple westerns would strengthen this point. Also, at 12 hours, the phospho-H3 signal suggests cells have entered mitosis. At this point USP9X and CEP131 abundance has not decreased, yet these proteins are not detectable in mitosis in the immunofluorescence images (Figure 1A). Does the abundance of these proteins decrease by western plot at later time points?

Authors: According to the reviewer's suggestion, a quantification of multiple westerns has been provided in Figure 2E and Figure S2C and the results indicate that USP9X and CEP131 in these cells change at a similar pace except 2 hours after synchronization, a time point at which the abundance of CEP131, but USP9X, dramatically increased. These results imply that CEP131 could be controlled by other factors at the initial S phase of the cell cycle. Accordingly, We have modified the relevant text.

Similar to CEP131¹, USP9X is redistributed and exported from centrosome (Figure 2B and Figure S2B), but the level of these proteins did not decrease, as demonstrated in Figure 2E and Figure S2D. To clearly display the centrosomal signal of USP9X, we

performed immunofluorescent assay using antibodies against USP9X in high dilution (1:500). Cytoplasmic staining of USP9X in mitotic cells was weak but could still be detected (Figure 2B and Figure S2B). Western blotting analysis indicated that the abundance of CEP131 and USP9X decreased at later time points, when cells left mitosis and proceeded into a new cycle (Figure S2D).

Reviewer #3: *Additionally, time of interaction between USP9X and CEP131 in Figure 2F does not seem to correlate with the initial increase in CEP131 abundance in Figure 2E.*

Authors: As stated earlier, CEP131 abundance could be regulated by other factors especially in initial S phase (two hours post synchronization) of the cell cycle.

Reviewer #3: *Page 12: Please provide biological replicates for the quantification in Figure 3D.*

Authors: The information has been provided in Figure 3D, Figure S3C, and Figure S3D.

Reviewer #3: *Does the loss of CEP131 from the centrosome specific in USP9X siRNA conditions? i.e. are other components also displaced (other than centrin)? Possibly other satellite components such as CEP290 or PCM1? The loss of CEP131 from centrosomes could represent a general loss of satellite components.*

Authors: As demonstrated in Figure S4A, the centrosomal localization of PCM1 was indeed mildly disrupted, while the centrosomal localization of CEP290 and Pericentrin was essentially unchanged in USP9X deficient cells.

Reviewer #3: *Figure 3F. Please provide quantification for this. The CEP131 abundance in MDA-MB-231, ECC1 and Ishkawa cells look reasonably similar while having noticeably different levels of USP9X. Perhaps a correlation plot would be more convincing.*

Authors: The quantification and correlation on these proteins have been provided in Figure 3G.

Reviewer #3: *Page 14: What is FLAG-tagged in Figure 4D? Is it FLAG-USP9X? Please indicate. The level of overexpression of the wt and C1566S forms of USP9X is not comparable. Is it possible to use a higher doxycycline concentration for the C1566S protein?*

Authors: FLAG-tagged USP9X is indicated in Figure 4D. Based on the reviewer's suggestion, we have used higher concentration of doxycycline to induce C1566S

expression and the corresponding deubiquitination assay has been re-performed as shown in Figure S5C.

Reviewer #3: *Page 15: The labels on Figure 4F are confusing. The bottom panel is labeled CEP131/K504, yet CEP131/K254R is indicated as being present in one of the lanes.*

Authors: Figure 4F has been carefully relabeled.

Reviewer #3: *Please include biological replicates for the quantification of Figure 4G.*

Authors: The information has been provided in Figure 4G and Figure S5F

Reviewer #3: *Page 16: Do the FACS profiles correspond to the cells analyzed in the immunofluorescence panels in Figure 5A and C? If so, the profiles look like mostly G1 cells. Can the authors provide A) a FACS profile of cycling cells to use for a comparison B) another method, such as cyclin A staining or western blot to demonstrate the cells are in the expected cell cycle stage.*

Authors: FACS profiles in original Figure 5A corresponded to unsynchronized cells, not cells analyzed in the immunofluorescence panels. In the revision, new FACS profiles and Western blotting analysis of the cells used in the immunofluorescence panels have been shown (Figure S6A), and FACS profiles of control and HU treated unsynchronized cells corresponding to the immunofluorescence panels have been provided in Figure 5C.

Reviewer #3: *For Figures 5B and C it would be nice to see western blots to judge protein abundance. For example, USP9X over expression should stabilize CEP131, but CEP131 siRNA should deplete the protein - it is necessary to judge what the final steady state level of the protein is in this case to interpret the functional data.*

Authors: Western blots have been provided in Figure 5B, Figure 5C, and other Figures as suggested.

Reviewer #3: *Page 18: Can Figure 6C be rearranged so all normal and all tumor lines are together?*

Authors: Since the samples we used were not paired from the same patients, to avoid confusing, we grouped the samples as shown in Figure 6C and run them on separate gels to display the different abundances of these proteins in breast cancer and normal breast tissues.

Reviewer #3: *Page 19: In Figure 7C, shRNA against CEP131 results in a weaker colony formation phenotype that using shRNA against USP9X. However, the abundance of CEP131 is lower using shRNA against CEP131 than in cells using USP9X shRNA. This is not entirely consistent with the model where USP9X regulates CEP131 stability.*

Authors: In our opinion, it is hard to compare the effect of USP9X depletion versus that of CEP131 depletion in Figure 7A and 7B, because: 1) the initial cell numbers used in live cell counting or colony formation assays were not the same; and 2) the efficiency of USP9X knockdown versus CEP131 knockdown would, inevitably, differ. Similarly, the discrepancy of the abundance of CEP131 in CEP131 knockdown cells versus USP9X knockdown cells could possibly come from the disparity of shRNA transfection/knockdown efficiency; 3) USP9X also targets substrates other than CEP131.

Reviewer #3: *Perhaps some of the data that is replicated in different cell lines can be presented in Supplementary information.*

Authors: To comply with the reviewer's suggestion, data in Figure 3A, Figure 3D, Figure 4C and Figure 5B have been moved to Supplementary information and shown as Figure S3A, Figure S3D, Figure S5B and Figure S6B, respectively.

Reviewer #3: *page7: excessive FLAG peptides: excess FLAG peptide(s) centriolar satellite protein CEP131 was also identified: were also identified*

Authors: These words have been corrected.

Reviewer #3: *Page 10: deubiquitinase US9X: USP9X*

Authors: The word has been corrected.

Reviewer #3: *Page 19: samples (Fig. 6D, upper panle) and: upper panel*

Authors: The typo has been corrected.

References for Reviewer #3

1. Staples CJ, *et al.* The centriolar satellite protein Cep131 is important for genome stability. *Journal of cell science* **125**, 4770-4779 (2012).
2. Mouchantaf R, Azakir BA, McPherson PS, Millard SM, Wood SA, Angers A.

The ubiquitin ligase itch is auto-ubiquitylated in vivo and in vitro but is protected from degradation by interacting with the deubiquitylating enzyme FAM/USP9X. *J Biol Chem* **281**, 38738-38747 (2006).

3. Perez-Mancera PA, *et al.* The deubiquitinase USP9X suppresses pancreatic ductal adenocarcinoma. *Nature* **486**, 266-270 (2012).
4. Schwickart M, *et al.* Deubiquitinase USP9X stabilizes MCL1 and promotes tumour cell survival. *Nature* **463**, 103-107 (2010).

Reviewers' comments:

Reviewer #1 (Remarks to the Author):

Review on the article "The X-linked Deubiquitinase USP9X Regulates Centrosome Duplication and Promotes Breast Carcinogenesis"

In this paper, USP9x is found to be a centrosome component that deubiquitinates its partner, CEP131, shown elsewhere to regulate centrosomal localisation of CDK2. USP9X and CEP131 are also overexpressed in breast cancer and it is suggested that USP9X promotes carcinogenesis through stabilizing CEP131.

The interaction between USP9X and CEP131 is very well supported by affinity purification and mass spectrometry, as well as by co-immunoprecipitation in cell lines. Importantly, the co-IP experiments on 4 different cell lines give consistent results and the interaction sites of USP9X and CEP131 are properly validated. The effect of USP9X overexpression on centrosome amplification and chromosome stability and the co-dependency of USP9X and CEP131 on these process are well demonstrated as is the mediation of this effect through CDK2 as previously shown.

In conclusion, this this is a valuable article but still requires some revision before publication.

Major points:

1) page 10: centrosomal and cytoplasmic localisation of USP9X

The immunofluorescence data presented suggests that Usp9X localisation is restricted to the centrosomes to the same extent as γ -tubulin, centrin, PCM1 or CEP131. This creates an illusion that this protein is localised exclusively at the centrosomes. However, Urbé et al, 2012 (FigS1C), Han et al, 2012 (Fig4) in HeLa cells, Mertz et al, 2015 (Fig5) in HEK293 cells show the extensive cytoplasmic distribution of Usp9X. The localisation of transgenic GFP-tagged USP9X in Fig2C of this article most likely does reflect true localisation to both centrosome and cytoplasm consistent with it having other partners involved in cell adhesion, Golgi function, apoptosis and trafficking (review in Murtaza et al, 2015). The authors should comment on this issue. However, further work is still required to show that the mainly cytoplasmic Usp9X relocates to the centrosome in G1/S. This requires localisation of USP9X by immunofluorescence in G1. It would be very surprising and unlikely, however, that all cytoplasmic Usp9X disappears in S/G2-M and relocates to the centrosomes. Since there is no indication, either in Results section or Methods, which anti-Usp9X antibody was used, we do not know how the authors' findings relate to the localisation in the above mentioned papers where a diffuse cytoplasmic localisation was reported.

2) page 14, line 17: rescue of CEP131 level by catalytically inactive USP9X

The authors state that "downregulation of CEP131 in these [Usp9X knockout] cells could be reverted by forced expression of wild-type USP9X, but not USP9X mutant (USP7/C1566S)". Here there is a minor problem - "USP7/C1566S" should be corrected to "USP9X/C1566S" (this seems to be a typo graphical error as USP7 is a totally different DUB). More importantly, according to FigureS5A the level of CEP131 seems to be restored almost to the wild type level when catalytically inactive USP9X mutant was overexpressed (compare lane 5 with lane 1 - WT - and lane 2-3 - Usp9X KO). Although the overexpression of wild type Usp9X gives a higher accumulation of CEP131 (lane 4), the authors should explain the rescue effect observed with the catalytically inactive USP9X, or soften their statement to "overexpression of USP9X mutant could only moderately revert the CEP131 downregulation".

3) poly-ubiquitin chain specificity of USP9X

The authors do not comment why they chose to investigate K29, K48, K63 polyubiquitin chains. It is particularly important, because Al-Hakim et al, 2008 demonstrated that Usp9X deubiquitinates only K29 and K33 poly-ubiquitylated NUA1 and MARK4 and is not able to hydrolyse K48 and K63 tetraubiquitin chains. According to McGouran et al, 2013 Usp9X (they call FAF-X) is able to process K11, K27, K29 di-ubiquitin probes, but shows only moderate activity to K33, K48, K63 di-ubiquitins. I think, the authors should offer an explanation because they state that K48 ubiquitin chains are the major forms opposed by Usp9X. According to the immunoblots they show in Figure4 and FigureS5, other types of polyubiquitins might be involved in CEP131 ubiquitination which can be opposed by Usp9X. This is because in FigureS5 the HA-Ub/wt signal seems to be orders of magnitude stronger than HA-Ub/K48-only. Ubiquitin chain type specific antibodies are available,

which can help reveal the exact nature of CEP131 ubiquitination and Usp9X specificity.

4) page 20: reference gene in Real-time qPCR experiments

Only one reference gene, namely β -actin, has been used in real-time qPCR experiments. I would like to draw the author's attention to the MIQE (minimum information for the publication of quantitative real-time PCR) guidelines published by Gemma Johnson et al. in 2014. According to these guidelines, when two samples differ less than 5-fold in RNA levels it is essential to use multiple reference genes. This is critical for the experiment at Figure 6B, where USP9X expression level was determined in normal and malignant tissue and where the average difference seems to be less than 5-fold. The use of beta-actin as a reference gene should be avoided, because it is well described that beta-actin expression is deregulated in many cancer types, including breast cancer and MCF-7 cells (reviewed in Guo et al, 2013).

5) page 24: centrosome amplification in tumors

It would be helpful if the authors could show the centrosome amplification in the tumors grown in athymic mice and the reduction of centrosome amplification upon Usp9X knock-down in these tumors. This could support their firm statement made at page 24, line 13: "USP9X promotes breast carcinogenesis, and that USP9X does so, through stabilizing the centriolar satellite protein CEP131". There are other USP9X substrates involved in carcinogenesis (reviewed in Murtaza et al., 2015), which were not investigated by the authors, and their involvement cannot be excluded. The overexpression of CEP131 just partially restores the growth of USP9x knocked-down tumors (Figure 7E), indicating that other pathways might be involved as well.

Minor remarks:

- the year of publication is missing at reference 57 in the reference list
- "in vitro ubiquitination assay" is mentioned at page 16, though it should be "in vitro deubiquitination assay" as it can be deduced from the figure legend and methods. This should be corrected because it is misleading.
- pg. 17, 1st line: "able to remove the ubiquitination" should be corrected to "able to remove ubiquitins".
- pg.17, 3rd line: "poly-ubiquitination chains conjugated onto CEP131" should be corrected to "poly-ubiquitin chains conjugated onto CEP131".

Reviewer #2 (Remarks to the Author):

The revised version of the manuscript: "The X-linked Deubiquitinase USP9X Regulates Centrosome Duplication and Promotes Breast Carcinogenesis" has been completely answering all of my questions and concern.

Reviewer #3 (Remarks to the Author):

The new experiments included in the revised version of the manuscript seem reasonable and well controlled. However, I still remain only partly convinced of their model and for this reason the lasting impact of this work remains unclear. I note the following issues:

1) The interpretation of the localization data aren't correct. PCM1 and CEP131 co-localize with centrin and show almost no satellite distribution. Although the is not a major issue, it is disturbing that they don't see proper satellites (in U2OS cells).

2) The evidence for USP9X regulating CEP131 is convincing, but unfortunately they also see effects on the stability of PCM1. The authors also observe a good correlation between PCM1 and USP9X in breast cancer cells (better than CEP131!). However, they basically ignore this new data. I think this complicates their interpretation and this needs to be fully integrated in their model.

3) USP9X is already known to affect cancer cells (oddly as both as an oncogene and tumour suppressor...). Therefore, the link between USP9X and CEP131 in the breast cancer model has to be air-tight since the novelty of USP9X is not really there. Additionally, the rescue experiments in this section of the manuscript are not very convincing which makes me question the biological validity of their interpretation.

Response to Reviewer #1's comments-

Major points:

Reviewer #1:

1) page 10: centrosomal and cytoplasmic localisation of USP9X

The immunofluorescence data presented suggests that Usp9X localisation is restricted to the centrosomes to the same extent as γ -tubulin, centrin, PCMI or CEP131. This creates an illusion that this protein is localised exclusively at the centrosomes. However, Urbé et al, 2012 (FigS1C), Han et al, 2012 (Fig4) in HeLa cells, Mertz et al, 2015 (Fig5) in HEK293 cells show the extensive cytoplasmic distribution of Usp9X. The localisation of transgenic GFP-tagged USP9X in Fig2C of this article most likely does reflect true localisation to both centrosome and cytoplasm consistent with it having other partners involved in cell adhesion, Golgi function, apoptosis and trafficking (review in Murtaza et al, 2015). The authors should comment on this issue. However, further work is still required to show that the mainly cytoplasmic Usp9X relocates to the centrosome in G1/S. This requires localisation of USP9X by immunofluorescence in G1. It would be very surprising and unlikely, however, that all cytoplasmic Usp9X disappears in S/G2-M and relocates to the centrosomes. Since there is no indication, either in Results section or Methods, which anti-Usp9X antibody was used, we do not know how the authors' findings relate to the localisation in the above mentioned papers where a diffuse cytoplasmic localisation was reported.

Authors: We thank for the reviewer's comments. To clearly display the centrosomal signal of USP9X, we performed immunofluorescent assays using antibodies against USP9X in high dilution (1:500). We believe that could be the reason why USP9X showed weak staining in the cytoplasm. To address the reviewer's concern, we have repeated the immunofluorescent assays with more concentrated antibodies against USP9X (1:200). Immunofluorescence analysis indicated that USP9X displayed extensive cytoplasmic, weak nuclear and evident centrosomal localization, the observation of which is consistent with the cellular distribution pattern of GFP tagged USP9X. The results have been provided in Figure 2A, Figure 2D, Figure 3F, Figure 4H and Figure S2A to replace the previous ones. Moreover, we demonstrated that the centrosomal signal of USP9X was severely abolished upon USP9X depletion (Figure 3F), indicating that the centrosomal staining of USP9X was not resulted from antibodies' non-specific binding.

In addition, more concentrated USP9X antibodies (1:200) were used to re-perform the

immunofluorescent assays with cells in different cell cycles. The new data in Figure 2B and Figure S2B indicated that in G₁, S and G₂ phases of the cell cycle USP9X displayed positive staining in both cytoplasm and centrosome, while the centrosomal localization of USP9X was largely diminished in metaphase.

The anti-USP9X antibody information (manufacture and catalogue No.) has been provided in Supplemental Methods.

Reviewer #1:

2) page 14, line 17: *rescue of CEP131 level by catalytically inactive USP9X*

The authors state that "downregulation of CEP131 in these [Usp9X knockout] cells could be reverted by forced expression of wild-type USP9X, but not USP9X mutant (USP7/C1566S)". Here there is a minor problem - "USP7/C1566S" should be corrected to "USP9X/C1566S" (this seems to be a typo graphical error as USP7 is a totally different DUB). More importantly, according to FigureS5A the level of CEP131 seems to be restored almost to the wild type level when catalytically inactive USP9X mutant was overexpressed (compare lane 5 with lane 1 - WT - and lane 2-3 - Usp9X KO). Although the overexpression of wild type Usp9X gives a higher accumulation of CEP131 (lane 4), the authors should explain the rescue effect observed with the catalytically inactive USP9X, or soften their statement to "overexpression of USP9X mutant could only moderately revert the CEP131 downregulation".

Authors: According to the reviewer's suggestion, we have changed the statement to "overexpression of USP9X mutant could only moderately revert the CEP131 downregulation". In addition, "USP7/C1566S" has been corrected to "USP9X/C1566S".

The rescue effect observed with the catalytically inactive USP9X may come from: 1) residual activity of the mutant *in vivo*, non-catalytic roles of USP9X or a combination of both; 2) excessive amounts of USP9X/C1566S might bind CEP131, blocking it from accessing its unidentified E3 ligase.

Reviewer #1:

3) *poly-ubiquitin chain specificity of USP9X*

The authors do not comment why they chose to investigate K29, K48, K63 polyubiquitin chains. It is particularly important, because Al-Hakim et al, 2008 demonstrated that Usp9X deubiquitinates only K29 and K33 poly-ubiquitylated NUA1 and MARK4 and is not able to hydrolyse K48 and K63 tetraubiquitin chains. According to McGouran et al, 2013 Usp9X (they call FAF-X) is able to process K11, K27, K29 di-ubiquitin probes, but shows only moderate activity to K33, K48, K63 di-ubiquitins. I think, the authors should offer an explanation because they state that K48 ubiquitin chains are the major forms opposed by Usp9X. According to the immunoblots they show in Figure4 and FigureS5 ,

other types of polyubiquitins might be involved in CEP131 ubiquitination which can be opposed by Usp9X. This is because in Figure S5 the HA-Ub/wt signal seems to be orders of magnitude stronger than HA-Ub/K48-only. Ubiquitin chain type specific antibodies are available, which can help reveal the exact nature of CEP131 ubiquitination and Usp9X specificity.

Authors: We appreciate the reviewer for this comment. Among the polyubiquitin chains, although K6, K27 and K33 polyubiquitin linkers have been reported to contribute to proteasomal targeting^{1, 2}, K11, K29 and K48 polyubiquitin conjugates are considered as the most relevant ones associated with proteasome degradation^{2, 3, 4, 5}, while K63 ubiquitin linkages are mainly involved in non-proteasomal pathways as a scaffolding modification in signal transduction¹. Given that the physical association of USP9X with CEP131 was detected primarily in S and G₂ phases of the cell cycle (Figure 2F), we did not investigate K11 ubiquitin linkages that are preferentially produced during mitosis and early G₁^{3, 6}. Thereby, we chose to investigate K29 and K48 polyubiquitin chains with K63 as a likely control.

Consistent with our observations that USP9X deubiquitinates K48 poly-ubiquitylated CEP131, USP9X has been reported to efficiently remove degradative K48-linked polyubiquitin chains on MCL1⁷ and XIAP⁸. To clearly reveal the exact nature of CEP131 ubiquitination and USP9X specificity, we performed deubiquitination assays with ubiquitin chain type specific antibodies and revealed that K48 ubiquitin chains on CEP131, but not K63 linkages, was opposed by USP9X (Figure S5E). This observation is in favor of our previous findings with ubiquitin mutants K48-only and K63-only. Considering that the preference of USP9X on different types of ubiquitin linkages has been reported, we assume that USP9X opposes specific ubiquitin linkages in a substrate or context dependent manner.

We agree with the reviewer's point that other types of polyubiquitins might be involved in CEP131 ubiquitination. It will be interesting to investigate whether CEP131 is subjected to K6, K11, K27 and K33 polyubiquitin conjugates, whether these types of ubiquitin linkages could be removed by USP9X, and how these modifications impact on the molecular behaviors and functionalities of CEP131.

All of the information has been incorporated into the Results or Discussion sections (paragraph 2, page 30) accordingly in the revision.

Reviewer #1:

4) page 20: reference gene in Real-time qPCR experiments

Only one reference gene, namely β -actin, has been used in real-time qPCR experiments.

I would like to draw the author's attention to the MIQE (minimum information for the

publication of quantitative real-time PCR) guidelines published by Gemma Johnson et al. in 2014. According to these guidelines, when two samples differ less than 5-fold in RNA levels it is essential to use multiple reference genes. This is critical for the experiment at Figure 6B, where USP9X expression level was determined in normal and malignant tissue and where the average difference seems to be less than 5-fold. The use of beta-actin as a reference gene should be avoided, because it is well described that beta-actin expression is deregulated in many cancer types, including breast cancer and MCF-7 cells (reviewed in Guo et al, 2013).

Authors: In responding to the reviewer's comments, we have re-performed quantitative real-time PCR to determine USP9X expression level in normal and malignant tissues with GAPDH and PUM1 (a normalizer used in breast cancer)^{9, 10} as reference genes. The results still hold and the new data has been provided in Figure 6B to replace the previous one with β -actin as a normalizer.

Reviewer #1:

5) page 24: centrosome amplification in tumors

It would be helpful if the authors could show the centrosome amplification in the tumors grown in athymic mice and the reduction of centrosome amplification upon Usp9X knock-down in these tumors. This could support their firm statement made at page 24, line 13: "USP9X promotes breast carcinogenesis, and that USP9X does so, through stabilizing the centriolar satellite protein CEP131". There are other USP9X substrates involved in carcinogenesis (reviewed in Murtaza et al., 2015), which were not investigated by the authors, and their involvement cannot be excluded. The overexpression of CEP131 just partially restores the growth of USP9x knocked-down tumors (Figure 7E), indicating that other pathways might be involved as well.

Authors: To comply with the reviewer's requests, we have examined the centrosome numbers in cultured xenograft tumors with immunofluorescent assays. Since puromycin resistant gene is carried by the integrated shRNA lentivirus, tumor cells were enriched by puromycin selection during culture post isolating by trypsinization. The results in Figure 7F indicated that mild centrosome amplification could be observed in control tumors, while the percentage of cells with centrosome amplification was indeed reduced in USP9X knocked-down tumors. Furthermore, we found that CEP131 gain of function overrode the effects induced by USP9X depletion (Figure 7F). These observations support the argument that USP9X promotes breast carcinogenesis through stabilizing the centriolar satellite protein CEP131.

We agree with the reviewer's point that other pathways might be involved in USP9X-promoted tumorigenesis and their involvement needs to be considered. To clarify this issue, we have already performed a series of experiments with other substrates of

USP9X: 1) In agreement with previous reports^{7, 11, 12, 13}, we found that USP9X knockdown did result in decreased expression of ITCH (Figure S10A), an E3 ligase involved in carcinogenesis^{12, 14}, and of MCL1, an anti-apoptotic regulator implicated in cancer^{15, 16, 17, 18}. However, USP9X, but not CEP131, could be co-immunoprecipitated by ITCH or MCL1 (Figure S10B); 2) Colony formation assays demonstrated that while overexpression of ITCH or MCL1 could rescue the growth inhibitory phenotype resulted from USP9X depletion to certain extent as CEP131 did, simultaneous expression of CEP131 and ITCH or MCL1 showed an additive effect (Figure S10C); 3) while loss of USP9X-associated defects of centrosome amplification could be rescued by CEP131 overexpression (Figure 5C and Figure S10D), overexpression of ITCH or MCL1 could not rescue the phenotype (Figure S10D); and 4) knockdown of either ITCH or MCL1 had no effect on centrosomal localization of USP9X and CEP131 (Figure S10E). These results suggest that CEP131 functions cooperatively with but independently of other USP9X substrates in USP9X-promoted breast cancer cell survival, and also provided an explanation for why overexpression of CEP131 only partially restores the growth of USP9X knocked-down tumors (paragraph 2, page 23).

Minor remarks:

Reviewer #1: *the year of publication is missing at reference 57 in the reference list.*

Authors: The missing information has been added in the revision and the reference is shown as No.22 in the revision.

Reviewer #1: *"in vitro ubiquitination assay" is mentioned at page 16, though it should be "in vitro deubiquitination assay" as it can be deduced from the figure legend and methods. This should be corrected because it is misleading.*

Authors: It has been corrected.

Reviewer #1: *pg. 17, 1st line: "able to remove the ubiquitination" should be corrected to "able to remove ubiquitins".*

Authors: It has been corrected.

Reviewer #1: *pg.17, 3rd line: "poly-ubiquitination chains conjugated onto CEP131" should be corrected to "poly-ubiquitin chains conjugated onto CEP131".*

Authors: It has been corrected.

Response to Reviewer #3's comments-

Reviewer #3:

1) The interpretation of the localization data aren't correct. PCM1 and CEP131 co-localize with centrin and show almost no satellite distribution. Although there is not a major issue, it is disturbing that they don't see proper satellites (in U2OS cells).

Authors: We appreciate the reviewer for this comment. In our initial investigation, we performed immunofluorescent assay using antibodies against PCM1 and CEP131 in high dilution (1:500 and 1:1000, respectively). We speculate that this could be the reason why satellite information of CEP131 and PCM1 was missed. To avoid confusion, we have repeated the immunofluorescent assays with more concentrated antibodies against PCM1 (1:200) and CEP131 (1:200). Immunofluorescence analysis indicated that PCM1 and CEP131 displayed evident satellite stainings, and the new data has been provided to replace the previous results in Figure 2C, Figure 2D, Figure 3F, Figure S4C and Figure S10E.

Reviewer #3:

2) The evidence for USP9X regulating CEP131 is convincing, but unfortunately they also see effects on the stability of PCM1. The authors also observe a good correlation between PCM1 and USP9X in breast cancer cells (better than CEP131!). However, they basically ignore this new data. I think this complicates their interpretation and this needs to be fully integrated in their model.

Authors: Indeed, we demonstrated that PCM1 is a potential substrate of USP9X (Figure S4A and S4B). Specifically, the centrosomal localization of PCM1 was mildly interrupted (Figure S4A), and the protein, but not mRNA, expression level of PCM1 decreased (Figure S4B) in USP9X deficient cells. Although we observed a better correlation between PCM1 and USP9X than that between CEP131 and USP9X in breast cancer samples, the effect of USP9X depletion on the expression of PCM1 was not as dramatic as that of CEP131 (Figure S4B), suggesting that PCM1 is a potential, but might not be a major substrate of USP9X in centrosome. These observations likely explain why CEP131 overexpression could not fully compensate centrosomal biogenesis defects induced by USP9X depletion.

Since the abundance and localization of PCM1 seem to be regulated by USP9X, we are curious that whether the regulation of CEP131 by USP9X is a secondary effect of

USP9X-promoted PCM1 stabilization. To test this, we performed the following experiments and found: 1) In USP9X deficient cells, FLAG-tagged CEP131 could be still effectively recruited to centrosome (Figure S4C), suggesting that mildly disrupted centrosomal localization of PCM1 associated with USP9X depletion has limited effect on CEP131 recruitment; 2) Severe loss of PCM1 indeed impaired CEP131 centrosomal localization as reported¹⁹, while mild loss of PCM1 (approximately 40 to 50 percentage left) failed to do so (Figure S4D); 3) The expression level of CEP131 was essentially not altered upon PCM1 knockdown (Figure S4D). Combining these observations and the findings that USP9X directly interacts with CEP131 and opposes its poly-ubiquitin linkages *in vitro*, we get the conclusion that the effect of USP9X on CEP131 stabilization is attributed to the interplay between these two molecules but not resulting from other potential USP9X centrosomal interactors such as PCM1, and USP9X-regulated PCM1 stabilization on centrosome activity, if it does so, seems to be independent of USP9X-promoted CEP131 stabilization. We believe that understanding how USP9X promotes PCM1 stabilization and whether/how PCM1 contributes to USP9X regulated centrosome biogenesis will be helpful in interpreting the function of USP9X in centrosome.

All of the information has been incorporated into the Results or Discussion sections (paragraph 1, page 29) accordingly in the revision.

Reviewer #3:

3) USP9X is already known to affect cancer cells (oddly as both as an oncogene and tumour suppressor...). Therefore, the link between USP9X and CEP131 in the breast cancer model has to be air-tight since the novelty of USP9X is not really there. Additionally, the rescue experiments in this section of the manuscript are not very convincing which makes me question the biological validity of their interpretation.

Authors: As the reviewer mentioned, the conflicting conclusions surrounding USP9X function as either a tumor suppressor or an oncogene indicate that USP9X plays a complex role in carcinogenesis. Specifically, several reports have portrayed USP9X as an oncogene in prostate cancer²⁰, lymphomas⁷ and colorectal carcinoma²¹, while two studies suggest that interfering with USP9X expression promotes a more rapid onset of PDAC (pancreatic ductal adenocarcinomas) in genetic mouse model^{12, 22}. However, a recent study pointed that USP9X is an oncogene in the context of established PDAC with xenograft model⁷, which is consistent with findings in other neoplasms. In this regard, USP9X may parallel the behavior of TGF- β observed in some cancers, where TGF- β behaves as a tumor suppressor during early stages, but as an oncogene in later stages^{23, 24}. Whether this paradigm is applicable for USP9X in tumorigenesis needs to be further characterized.

To further establish the link between USP9X and CEP131 in breast carcinogenesis, we isolated breast cancer cells from frozen xenograft tumors, cultured the cells under puromycin selection (puromycin resistant gene together with shRNA cassette carried by the shRNA expressing lentivirus has been integrated into the genome of tumor cells) and examined the centrosome numbers with immunofluorescent assays. The results in Figure 7F indicated that mild centrosome amplification could be observed in control tumors, while the percentage of cells with centrosome amplification was indeed reduced in USP9X knocked-down tumors. Furthermore, we found that CEP131 gain of function overrode the effects induced by USP9X depletion. These observations further strengthen the argument that USP9X-promoted CEP131 stabilization plays a role of importance in breast carcinogenesis.

The novelty of our work is based on the weight of following observations: 1) although USP9X has been implicated in several pathological states including various malignancies and centrosome-associated CEP family proteins are considered to be potent tumor suppressors or oncogenes, mechanistic insights into the role of USP9X and CEP proteins in cancer development and progression, in particular, breast carcinogenesis, remain to be investigated; and 2) defective centrosome duplication is implicated in microcephaly and primordial dwarfism as well as various ciliopathies and cancers, yet how the centrosome biogenesis is regulated remains poorly understood. Through a body of work, we first discovered that USP9X is an integral component of the centrosome and serves to specifically stabilize the centriolar satellite protein CEP131. Second, we revealed that CEP131-regulated centrosomal localization of CDK2, a cyclin-dependent kinase with an established role in centrosome duplication, is required for USP9X-promoted centrosome biogenesis. Third, we demonstrated that USP9X, in doing so, impacts on chromosome stability and eventually promotes breast carcinogenesis.

Unlike numerous E3 ligase encoded by ~600 genes in human cells, there are only approximately 95 putative DUBs²⁵. Thus, each DUB is likely to oppose ubiquitin linkages of multiple proteins. Specifically, deubiquitinase USP7 has been reported to play important roles in tumorigenesis through removing ubiquitin species from tumor suppressors p53^{26, 27}, PTEN^{28, 29}, FOXO³⁰ and claspin³¹ or oncogenes MDM2^{27, 32}, PHF8³³ and UHRF1^{34, 35, 36}. We envision the same paradigm could be true for USP9X, in which USP9X targets multiple substrates involved in distinct or even converged signaling pathways and eventually impacts on tumorigenesis. This likely explains why expression of CEP131 only partially rescued the phenotype of depleting USP9X.

To test the above hypothesis, we have performed a series of experiments with other substrates of USP9X: 1) In agreement with previous reports^{7, 11, 12, 13}, we found that USP9X knockdown did result in decreased expression of ITCH (Figure S10A), an E3 ligase involved in carcinogenesis^{12, 14}, and of MCL1, an anti-apoptotic regulator

implicated in cancer^{15, 16, 17, 18}, However, USP9X, but not CEP131, could be co-immunoprecipitated by ITCH or MCL1 (Figure S10B); 2) Colony formation assays demonstrated that while overexpression of ITCH or MCL1 could rescue the growth inhibitory phenotype resulted from USP9X depletion to certain extent as CEP131 did, simultaneous expression of CEP131 and ITCH or MCL1 showed an additive effect (Figure S10C); 3) while loss of USP9X-associated defects of centrosome amplification could be rescued by CEP131 overexpression (Figure 5C and Figure S10D), overexpression of ITCH or MCL1 could not rescue the phenotype (Figure S10D); and 4) knockdown of either ITCH or MCL1 had no effect on centrosomal localization of USP9X and CEP131 (Figure S10E). These results suggest that CEP131 functions cooperatively with but independently of other USP9X substrates in USP9X-promoted breast cancer cell survival, and also provide an explanation for why overexpression of CEP131 could not fully restore the growth of USP9X knocked-down tumors (paragraph 2, page 23).

References

1. Yau R, Rape M. The increasing complexity of the ubiquitin code. *Nature cell biology* **18**, 579-586 (2016).
2. Xu P, *et al.* Quantitative proteomics reveals the function of unconventional ubiquitin chains in proteasomal degradation. *Cell* **137**, 133-145 (2009).
3. Meyer HJ, Rape M. Enhanced protein degradation by branched ubiquitin chains. *Cell* **157**, 910-921 (2014).
4. Miranda M, Sorkin A. Regulation of receptors and transporters by ubiquitination: new insights into surprisingly similar mechanisms. *Molecular interventions* **7**, 157-167 (2007).
5. Kim W, *et al.* Systematic and quantitative assessment of the ubiquitin-modified proteome. *Molecular cell* **44**, 325-340 (2011).
6. Matsumoto ML, *et al.* K11-linked polyubiquitination in cell cycle control revealed by a K11 linkage-specific antibody. *Molecular cell* **39**, 477-484 (2010).
7. Schwickart M, *et al.* Deubiquitinase USP9X stabilizes MCL1 and promotes tumour cell survival. *Nature* **463**, 103-107 (2010).
8. Engel K, *et al.* USP9X stabilizes XIAP to regulate mitotic cell death and

- chemoresistance in aggressive B-cell lymphoma. *EMBO molecular medicine* **8**, 851-862 (2016).
9. Lyng MB, Laenkholtm AV, Pallisgaard N, Ditzel HJ. Identification of genes for normalization of real-time RT-PCR data in breast carcinomas. *BMC cancer* **8**, 20 (2008).
 10. Tilli TM, Castro Cda S, Tuszynski JA, Carels N. A strategy to identify housekeeping genes suitable for analysis in breast cancer diseases. *BMC genomics* **17**, 639 (2016).
 11. Mouchantaf R, Azakir BA, McPherson PS, Millard SM, Wood SA, Angers A. The ubiquitin ligase itch is auto-ubiquitylated in vivo and in vitro but is protected from degradation by interacting with the deubiquitylating enzyme FAM/USP9X. *The Journal of biological chemistry* **281**, 38738-38747 (2006).
 12. Perez-Mancera PA, *et al.* The deubiquitinase USP9X suppresses pancreatic ductal adenocarcinoma. *Nature* **486**, 266-270 (2012).
 13. Trivigno D, Essmann F, Huber SM, Rudner J. Deubiquitinase USP9x confers radioresistance through stabilization of Mcl-1. *Neoplasia* **14**, 893-904 (2012).
 14. Salah Z, Itzhaki E, Aqeilan RI. The ubiquitin E3 ligase ITCH enhances breast tumor progression by inhibiting the Hippo tumor suppressor pathway. *Oncotarget* **5**, 10886-10900 (2014).
 15. Gao J, *et al.* MiR-26a inhibits proliferation and migration of breast cancer through repression of MCL-1. *PloS one* **8**, e65138 (2013).
 16. Inuzuka H, *et al.* SCF(FBW7) regulates cellular apoptosis by targeting MCL1 for ubiquitylation and destruction. *Nature* **471**, 104-109 (2011).
 17. Wertz IE, *et al.* Sensitivity to antitubulin chemotherapeutics is regulated by MCL1 and FBW7. *Nature* **471**, 110-114 (2011).
 18. Yang L, *et al.* Wnt modulates MCL1 to control cell survival in triple negative breast cancer. *BMC cancer* **14**, 124 (2014).
 19. Staples CJ, *et al.* The centriolar satellite protein Cep131 is important for genome stability. *Journal of cell science* **125**, 4770-4779 (2012).

20. Wang S, *et al.* Ablation of the oncogenic transcription factor ERG by deubiquitinase inhibition in prostate cancer. *Proceedings of the National Academy of Sciences of the United States of America* **111**, 4251-4256 (2014).
21. Harris DR, Mims A, Bunz F. Genetic disruption of USP9X sensitizes colorectal cancer cells to 5-fluorouracil. *Cancer biology & therapy* **13**, 1319-1324 (2012).
22. Mann KM, *et al.* Sleeping Beauty mutagenesis reveals cooperating mutations and pathways in pancreatic adenocarcinoma. *Proceedings of the National Academy of Sciences of the United States of America* **109**, 5934-5941 (2012).
23. Ikushima H, Miyazono K. TGFbeta signalling: a complex web in cancer progression. *Nature reviews Cancer* **10**, 415-424 (2010).
24. Bierie B, Moses HL. Tumour microenvironment: TGFbeta: the molecular Jekyll and Hyde of cancer. *Nature reviews Cancer* **6**, 506-520 (2006).
25. Komander D, Clague MJ, Urbe S. Breaking the chains: structure and function of the deubiquitinases. *Nature reviews Molecular cell biology* **10**, 550-563 (2009).
26. Epping MT, Meijer LA, Krijgsman O, Bos JL, Pandolfi PP, Bernards R. TSPYL5 suppresses p53 levels and function by physical interaction with USP7. *Nature cell biology* **13**, 102-108 (2011).
27. Sheng Y, *et al.* Molecular recognition of p53 and MDM2 by USP7/HAUSP. *Nature structural & molecular biology* **13**, 285-291 (2006).
28. Morotti A, *et al.* BCR-ABL disrupts PTEN nuclear-cytoplasmic shuttling through phosphorylation-dependent activation of HAUSP. *Leukemia* **28**, 1326-1333 (2014).
29. Song MS, *et al.* The deubiquitylation and localization of PTEN are regulated by a HAUSP-PML network. *Nature* **455**, 813-817 (2008).
30. van der Horst A, *et al.* FOXO4 transcriptional activity is regulated by monoubiquitination and USP7/HAUSP. *Nature cell biology* **8**, 1064-1073 (2006).
31. Faustrup H, Bekker-Jensen S, Bartek J, Lukas J, Mailand N. USP7 counteracts SCFbetaTrCP- but not APCdh1-mediated proteolysis of Claspin. *The Journal of cell biology* **184**, 13-19 (2009).

32. Hu M, Gu L, Li M, Jeffrey PD, Gu W, Shi Y. Structural basis of competitive recognition of p53 and MDM2 by HAUSP/USP7: implications for the regulation of the p53-MDM2 pathway. *PLoS biology* **4**, e27 (2006).
33. Wang Q, *et al.* Stabilization of histone demethylase PHF8 by USP7 promotes breast carcinogenesis. *The Journal of clinical investigation* **126**, 2205-2220 (2016).
34. Ma H, *et al.* M phase phosphorylation of the epigenetic regulator UHRF1 regulates its physical association with the deubiquitylase USP7 and stability. *Proceedings of the National Academy of Sciences of the United States of America* **109**, 4828-4833 (2012).
35. Felle M, *et al.* The USP7/Dnmt1 complex stimulates the DNA methylation activity of Dnmt1 and regulates the stability of UHRF1. *Nucleic acids research* **39**, 8355-8365 (2011).
36. Mudbhary R, *et al.* UHRF1 overexpression drives DNA hypomethylation and hepatocellular carcinoma. *Cancer cell* **25**, 196-209 (2014).

REVIEWERS' COMMENTS:

Reviewer #1 (Remarks to the Author):

Reaction to the "The X-linked Deubiquitinase USP9X Regulates Centrosome Duplication and Promotes Breast Carcinogenesis. " Nat Comm paper review rebuttal

Major remarks rebuttal

1) I accept the authors reply to this question. By increasing the concentration of the primary antibody they managed to reveal the non-centrosomal pool of Usp9X and they show by RNAi that the cytoplasmic signal is specific.

2) I accept the authors response. They corrected the minor typographical error (Usp7 instead of Usp9X). They also softened the statement "Usp9x catalytic mutant is not able to restore the CEP131 level in Usp9X mutant " to "overexpression of USP9X mutant (USP9X/C1566S) could only moderately revert the CEP131 downregulation". Thi is now in agreement with Figure5S and explains the partial rescue by the non-catalytic function of Usp9X or by overexpression.

3) The authors provide an excellent response to this question. They have not just explained and commented upon the K48 specificity of Usp9X but have performed additional, very convincing experiments to further confirm their finding. I appreciate this very much.

4) I accept the authors response to this question. According to my suggestion they redid the real-time experiments with reference genes other than beta-actin. These reference genes are GAPDH and PUM1 which are published as known breast cancer reference genes (as cited). Although their findings are in accord with their previous (beta-actin normalised) measurements, I definitely believe that their presentation of this qPCR data is much more convincing.

5) I have ambiguous feelings about the images. I really appreciate the authors effort to do experiments to demonstrate centrosome amplification upon Usp9X downregulation and also to demonstrate that CEP131 functions cooperatively with, but independently of, other USP9X substrates in USP9X-promoted breast cancer cell survival. The latter is convincing but the demonstration of the first part, the centrosome amplification is less so. On Figure7F the images showing amplified (and presumably clustered) centrosomes are very blurry. I would recommend the replacement of the image with a higher resolution but less exposed image, where the individual centrosomes could be seen. However, since the experiments and quantification are convincing (especially together with the other results), I am happy to accept this response.

Minor remarks rebuttal:

All the minor corrections were performed in accordance with our requests

In summary, I believe the authors have satisfactorily addressed all comments.

Reviewer #3 (Remarks to the Author):

Overall, the authors have identified the centriolar satellite protein CEP131 as an interacting protein of USP9X. They provide evidence that USP9X deubiquitinates CEP131 resulting in the stabilization of the protein. In turn, CEP131 promotes centriole over-duplication. Moreover, the over-expression of USP9X in breast cancer cells is correlated with increased expression of CEP131. The depletion of USP9X and CEP131 from mouse tumor models reduces tumor growth and this can be partially rescued by re-expression of CEP131.

The authors have addressed all concerns raised during the previous round of review. We have the following minor points:

1. Given the effect of USP9X loss on CEP131 protein abundance in other instances in the manuscript (see Fig. 3A and B), the change in CEP131 abundance after USP9X depletion in Figure 7B is unconvincing. Can the authors supply an alternative western blot for this experiment?

2.p 26. The authors state 'Fluorescent microscopic analysis revealed that in S/G2 phases of the

cell cycle, USP9X is spatially restricted to centrosome'. The authors demonstrate that UPS9X localized to the centrosome in G1 cells, and additionally, that it also resides in the cytoplasm during all cell cycle stages. Please soften statement to reflect these data.

3. p29. 'suggesting that PCM1 is a potential, but might not be a major substrate of USP9X in centrosome.' This is an incomplete sentence. Perhaps 'suggesting the PCM1 is a potential, but not major, substrate of USP9X at the centrosome'.

4. p30 'we speculate that CEP131 overexpression might result in expansion of PCM cloud thus build enough space for centriolar assembly'. I am unsure of how an expanded PCM cloud is required for centriolar assembly. Perhaps the authors can expand slightly on this idea or omit it.

5. The authors should re-iterate in their discussion that the link between CEP131 and USP9X is one of multiple pathways that appear to act in cancer cells. As demonstrated in Figure S10C, the simultaneous expression of multiple USP9X targets rescue MCF-7 colony formation in an additive manner. This is an important point that puts USP9X at the apex of a regulation network that affects multiple cellular processes.

Response to Reviewer #1's comments-

Major points:

Reviewer #1:

5) I have ambiguous feelings about the images. I really appreciate the authors' effort to do experiments to demonstrate centrosome amplification upon Usp9X downregulation and also to demonstrate that CEP131 functions cooperatively with, but independently of, other USP9X substrates in USP9X-promoted breast cancer cell survival. The latter is convincing but the demonstration of the first part, the centrosome amplification is less so. On Figure7F the images showing amplified (and presumably clustered) centrosomes are very blurry. I would recommend the replacement of the image with a higher resolution but less exposed image, where the individual centrosomes could be seen. However, since the experiments and quantification are convincing (especially together with the other results), I am happy to accept this response.

Authors: To comply with the reviewer's comment, we have re-captured images from immunofluorescent stainings with cultured xenografts. The high resolution and less exposed images have been provided in Fig. 7f to replace the previous ones.

Response to Reviewer #3's comments-

Reviewer #3:

1). *Given the effect of USP9X loss on CEP131 protein abundance in other instances in the manuscript (see Fig. 3A and B), the change in CEP131 abundance after USP9X depletion in Figure 7B is unconvincing. Can the authors supply an alternative western blot for this experiment?*

Authors: An alternative western blot has been provided in Fig. 7b.

Reviewer #3:

2). *p 26. The authors state 'Fluorescent microscopic analysis revealed that in S/G₂ phases of the cell cycle, USP9X is spatially restricted to centrosome'. The authors demonstrate that UPS9X localized to the centrosome in G₁ cells, and additionally, that it also resides in the cytoplasm during all cell cycle stages. Please soften statement to reflect these data.*

Authors: The statement has been softened as "Immunofluorescent microscopy analysis revealed that in G₁/S/G₂ phases of the cell cycle, USP9X is distributed not only in cytoplasm but also in centrosome" (paragraph 1, page 21).

Reviewer #3:

3). *p29. 'suggesting that PCMI is a potential, but might not be a major substrate of USP9X in centrosome.' This is an incomplete sentence. Perhaps 'suggesting the PCMI is a potential, but not major, substrate of USP9X at the centrosome'.*

Authors: This sentence has been rewritten according to the reviewer's suggestion (paragraph 1, page 23).

Reviewer #3:

4). *p30 'we speculate that CEP131 overexpression might result in expansion of PCM cloud thus build enough space for centriolar assembly'. I am unsure of how an expanded PCM cloud is required for centriolar assembly. Perhaps the authors can expand slightly on this idea or omit it.*

Authors: This sentence has been deleted (paragraph 2, page 23).

Reviewer #3:

5). The authors should re-iterate in their discussion that the link between CEP131 and USP9X is one of multiple pathways that appear to act in cancer cells. As demonstrated in Figure S10C, the simultaneous expression of multiple USP9X targets rescue MCF-7 colony formation in an additive manner. This is an important point that puts USP9X at the apex of a regulation network that affects multiple cellular processes.

Authors: We appreciate the reviewer for this comment. We have incorporated the discussion into the last paragraph of page 24.